# GALE: GRADIENT ACTIVATION LOW-RANK EXTRACTION FOR FAST MEMORY EFFICIENT LARGE LANGUAGE MODEL TRAINING

## ABSTRACT

Training large language models (LLMs) is resource-intensive, with optimizer states consuming a significant portion of GPU memory. While system-level optimizations like ZeRO and new hardware have mitigated this at scale, memory constraints remain a critical hurdle for democratizing LLM training on consumer-grade hardware and maximizing efficiency on constrained clusters. Current memory-saving strategies involve a trade-off: Parameter-Efficient Fine-Tuning (PEFT) methods are fast and memory-efficient but often underperform compared to full-parameter training. Conversely, gradient projection methods like GaLore enable full-parameter learning with a low memory footprint, though the computational cost of Singular Value Decomposition (SVD) remains a bottleneck. While recent works have attempted to mitigate this, we introduce **Gradient Activation Low-rank Extraction (GALE)**, a method that advances this optimization further. GALE re-engineers the gradient projection pipeline. Instead of SVD, it uses a randomized sketching + QR decomposition algorithm. This approach eliminates a key computational bottleneck in the update step by accelerating the low-rank optimizer update step by up to $23\times$ over GaLore. This removes the overhead of gradient projection, resulting modest gains in training throughput. We present GALE in several variants, including an optimized version using mixed-precision fused kernels, which both modestly improve throughput and boost final task performance. When pre-training LLaMA models on the C4 dataset, GALE's task performance maintains that of GaLore while consistently and substantially outperforming PEFT methods. On the GLUE fine-tuning benchmark, GALE reduces the performance gap to leading PEFT techniques while removing the optimizer overhead of GaLore, thereby achieving higher training throughput than prior gradient projection methods and making full-parameter fine-tuning more computationally practical. By effectively balancing memory, performance, and computational speed, GALE sets a new practical frontier for efficient full-parameter LLM training. Code to replicate our findings can be found at GitHub.

## 1 INTRODUCTION

The ever-growing scale of Large Language Models (LLMs) has led to prohibitive memory and computational costs (Dao et al., 2022). Standard mixed-precision training of a 7B-parameter model with AdamW requires 80–100 GB of GPU memory (including optimizer states, gradients, and activations), creating barriers without complex system-level optimizations (e.g., ZeRO-3) or specialized hardware. This resource barrier concentrates cutting-edge LLM development within well-resourced labs, creating urgent need for memory-efficient training solutions.

Two main technique families have emerged. **Parameter-Efficient Fine-Tuning (PEFT)** freezes most pre-trained weights and updates only a small parameter subset. Examples include **LoRA** (Hu et al., 2022), which injects trainable low-rank matrices, **Prefix-Tuning** (Li & Liang, 2021), which optimizes task-specific vectors, and **(IA)**[3] (Liu et al., 2022), which rescales internal activations. While exceptionally fast and memory-light, PEFT constrains optimization to a fixed low-rank subspace, often creating a performance gap versus full-rank training.

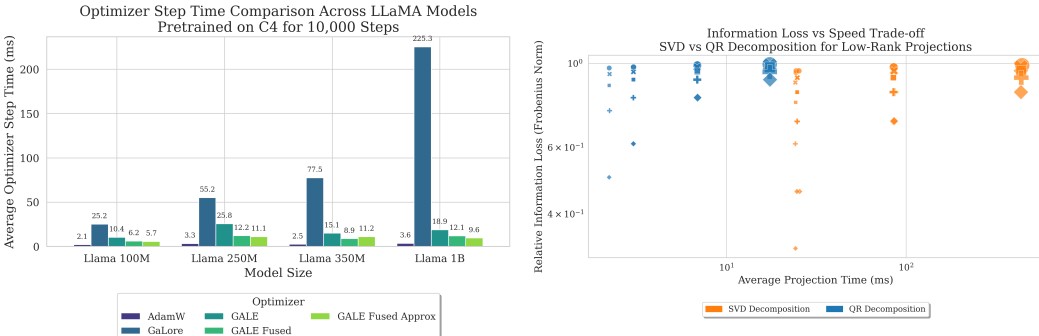

Figure 1: **(Left)** Comparison of average optimizer update step time (ms) vs Model size for full-rank AdamW baseline, GaLore and GALE variants. GALE's update step is up to 23x faster than GaLore. **(Right)** Tradeoff between information loss and projection speed of SVD vs QR Decomposition for various matrix sizes. QR decomposition offers significantly faster projection for comparable information loss, motivating our approach.

**Gradient projection** offers a powerful alternative. **GaLore** (Zhao et al., 2024) demonstrated breakthrough results, enabling full-parameter training with PEFT-comparable memory footprints. It successfully trained a 7B LLM on a single consumer GPU with 24GB VRAM, achieving higher task performance than popular PEFT techniques. GaLore projects the full gradient matrix into a low-rank subspace *before* the optimizer step, maintaining optimizer states in dramatically smaller dimensions. However, GaLore introduces a critical bottleneck: while its SVD-based projection recomputes only at specific intervals (e.g., every 200 or 1000 steps), these steps are $40\times$ slower than standard steps. This infrequent but heavy overhead significantly reduces overall training throughput (see Figure 1), stemming from reliance on exact, full-rank Singular Value Decomposition (SVD).

GALE fundamentally re-engineers the gradient projection pipeline around this insight. We present three variants: **GALE (Native)**, the foundational randomized QR projection pipeline in pure PyTorch, demonstrating significant speedup over SVD-based approaches; **GALE Fused**, which leverages custom mixed-precision CUDA kernels to accelerate optimizer updates, further improving throughput and task performance; and **GALE Fused Approx**, which uses subsampled sketching to speed up projection itself, creating a controllable tradeoff between throughput and projection accuracy.

Our contributions are:

- We analyze the SVD computational bottleneck in GaLore and demonstrate that our approach offers a superior solution with $23\times$ higher optimizer step times compared to GaLore, which is an advancement over existing SOTA in gradient projection methods for the same task performance
- We demonstrate that GALE is complimentary to other memory efficient optimizers like AdamW8Bit and Adafactor, while also having synergy with quantized projection methods like Q-GaLore
- We release our code and pre-trained model checkpoints to facilitate future research and make efficient LLM training more accessible.

## 2 RELATED WORKS

### 2.1 LOW-RANK TRAINING AND ADAPTATION

The use of low-rank structures in neural networks has evolved significantly. Early methods focused on training models with low-rank factorized weights from scratch (Sainath et al., 2013). More recently, the PEFT paradigm has become dominant for model adaptation. **LoRA** (Hu et al., 2022) is the most prominent example, which freezes pre-trained weights and injects trainable low-rank matrices ($W = W_0 + BA$).

A more recent and impactful alternative to LoRA is DoRA (Liu et al., 2024), which decomposes pre-trained weights into magnitude and direction, applying low-rank adaptation specifically to directions. DoRA improves the learning capacity and stability of LoRA without added inference overhead, consistently outperforming LoRA on challenging tasks. VeRA (Kopiczko et al.) further reduces trainable parameters by sharing low-rank matrices across layers and learning scaling vectors, demonstrating strong performance with a much smaller memory footprint compared to LoRA and DoRA.

Orthogonal to these approaches, FourierFT (Gao et al., 2024) takes a frequency domain view, representing parameter updates in the spectral domain and learning only a few spectral coefficients per layer. This achieves comparable or better performance than LoRA and DoRA, while requiring far fewer trainable parameters.

## 2.2 MEMORY-EFFICIENT OPTIMIZATION

Several techniques directly target the memory cost of the optimizer. **8-bit optimizers** (Dettmers et al.), for example, quantize optimizer states to reduce their footprint from 32-bits to 8-bits. **Adafactor** (Shazeer & Stern, 2018) reduces memory by factorizing the second-moment matrix of the Adam optimizer. Other methods fuse the backward pass with the optimizer step, which avoids storing the full gradient tensor in memory (Lewandowski & Kosson, 2023). GALE is a gradient-focused method that is complementary to these optimizer-focused approaches and can potentially be combined with them for even greater memory savings.

## 2.3 GRADIENT PROJECTION METHODS

GALE builds upon **subspace learning**, where weight updates are constrained to a low-dimensional subspace. **GaLore** (Zhao et al., 2024) introduced this for full-parameter training by projecting gradients into a low-rank form ($P^T G Q$) to save optimizer memory. However, its reliance on exact Singular Value Decomposition (SVD) creates a significant computational bottleneck.

Recent methods address this limitation. **Q-GaLore** (Zhang et al.) combines low-rank projection with INT4 quantization to lower memory usage, though quantization overheads can limit training throughput. **GaLore 2** (Su et al., 2025) targets the SVD bottleneck via randomized SVD (Halko et al., 2011), while **APOLLO** (Zhu et al.) replaces AdamW states with low-rank learning rate scaling via random projections to match SGD memory costs.

GALE advances this line of research by maximizing projection throughput. While GaLore 2 speeds up SVD through iterative sketching and reconstruction, GALE bypasses these steps using a direct randomized QR approach. We demonstrate that the orthonormal basis from the sketched gradient is sufficient for high-performance training, eliminating the need for exact singular vectors. By combining this simplified pipeline with our Fused and Fused Approx kernels, GALE outperforms both the quantization-heavy Q-GaLore and APOLLO's approximated scaling, establishing a new Pareto frontier for wall-clock training efficiency.

## 3 METHODOLOGY

The primary memory bottleneck in training large models stems from optimizer states, which typically require storing first and second moments ($M_t, V_t$) for every trainable weight matrix $W_t \in \mathbb{R}^{m \times n}$. GALE is designed to enable full-parameter learning with a minimal memory footprint by re-engineering the gradient projection pipeline for high computational throughput. Instead of maintaining optimizer states in the full $m \times n$ dimension, GALE projects the gradient into a low-rank subspace and performs the optimizer update in this compressed representation. The core of GALE is a fast, randomized two-stage algorithm that replaces the expensive SVD decomposition used in prior work.

## 3.1 THE GALE PIPELINE: RANDOMIZED SUBSPACE PROJECTION

The central innovation of GALE is its method for identifying the projection basis. It uses a fast, randomized two-stage procedure based on principles from randomized numerical linear algebra (Halko et al., 2011) to efficiently find a high-quality basis for the gradient's dominant subspace.

**Stage 1: Subspace Sketching.** First, we create a low-dimensional sketch, $Y$, that captures the dominant spectral information of the gradient matrix $G_t \in \mathbb{R}^{m \times n}$. This is achieved by multiplying the gradient by a random Gaussian matrix $\Omega$. To handle matrices of any shape efficiently, we operate on the smaller of the two dimensions. The sketch $Y$ is formed as follows:

$$Y = \begin{cases} G_t \Omega, & \Omega \in \mathbb{R}^{n \times r_{os}} & \text{if } m \geq n \\ G_t^T \Omega, & \Omega \in \mathbb{R}^{m \times r_{os}} & \text{if } m < n \end{cases} \tag{1}$$

The column space of the resulting small matrix $Y$ serves as a proxy for the top-$r$ subspace of $G_t$. To improve the quality of this approximation, we use oversampling, setting the sketch rank to $r_{os} = \max(r \cdot f_{os}, r+10)$ with a typical oversampling factor $f_{os} = 2$. For numerical stability, this sketching computation is performed in 32-bit floating-point precision to preserve the geometric structure of the subspace against potential precision loss.

**Stage 2: Orthonormal Basis Extraction.** Next, we apply a thin QR decomposition to the small sketch matrix $Y$ to find an orthonormal basis for its range: $Y = QR_{\text{sketch}}$. This factorization is a direct, non-iterative algorithm, making it significantly faster than SVD. The first $r$ columns of the resulting orthogonal matrix $Q_{\text{sketch}}$ form our final projection matrix, denoted as $Q \in \mathbb{R}^{n \times r}$ if $m \geq n$ or $P \in \mathbb{R}^{m \times r}$ if $m < n$.

**Optimizer Update in Low-Rank Subspace.** Once the projection matrix is identified, the full gradient $G_t$ is projected into the low-rank subspace ($G_{\text{proj}} = G_t Q$ or $G_{\text{proj}} = P^T G_t$). The optimizer states are maintained for this much smaller matrix, and the update is performed in this compressed space. The resulting low-rank weight update, $\Delta W_{\text{proj}}$, is then projected back to the full dimension ($\Delta W_t = \Delta W_{\text{proj}} Q^T$ or $\Delta W_t = P \Delta W_{\text{proj}}$) and applied to the model weights.

## 3.2 ALGORITHMIC IMPLEMENTATION AND PERFORMANCE VARIANTS

The complete GALE training procedure is detailed in Algorithm 1. Building on this foundation, we introduce specialized variants to further optimize performance.

**GALE (Native).** This is the foundational PyTorch implementation of our randomized QR projection pipeline, serving as the baseline to demonstrate the speedup of randomized subspace identification over exact SVD.

**GALE Fused.** This variant optimizes the optimizer update step (Lines 20 and 26 in Algorithm 1). We implement a custom fused CUDA kernel that performs the entire AdamW update on the low-rank tensors ($G_{\text{proj}}, M_{t-1}, V_{t-1}$) in a single call. This fusion reduces kernel launch overhead and improves GPU memory bandwidth utilization (Dao et al., 2022). The kernel uses a mixed-precision scheme: it loads 'bfloat16' tensors, converts them to 'float32' for numerically stable internal arithmetic, and writes the updated 'bfloat16' states back to memory, following modern best practices (Micikevicius et al., 2018).

**GALE Fused Approx.** For maximum throughput, this variant accelerates the subspace identification step itself (Lines 8–15 in Algorithm 1). After computing the sketch matrix $Y \in \mathbb{R}^{d \times r_{os}}$, we form an approximate sketch $Y_{\text{approx}}$ by randomly selecting a subset of its columns. The number of columns selected is $c = \max(r, \lfloor r_{os} \cdot f_{\text{approx}} \rfloor)$, where $f_{\text{approx}} \in (0, 1]$ is a tunable approximation factor. The QR decomposition is then performed on this smaller $Y_{\text{approx}}$. This subsampled QR technique further reduces the latency of finding the projection basis, creating a controllable trade-off between wall-clock speed and projection accuracy. This variant also utilizes the fused CUDA kernel for the optimizer step.

---

**Algorithm 1** The GALE Training Algorithm

---

1: **Input:** Parameters $W$, rank $r$, learning rate $\eta$, update interval $k_{\text{upd}}$, oversampling factor $f_{os}$
2: Initialize low-rank optimizer states $M_0, V_0 \leftarrow 0$
3: **for** each training step $t = 1, 2, \ldots$ **do**
4:     Compute full gradient $G_t \leftarrow \nabla_{W_{t-1}} \mathcal{L}$
5:     **if** $(t-1) \pmod{k_{\text{upd}}} == 0$ **then**
        {Refresh projection basis periodically}
6:         Let $G_t \in \mathbb{R}^{m \times n}$
7:         Set oversampling rank $r_{os} \leftarrow \max(r \cdot f_{os}, r + 10)$
8:         **if** $m \geq n$ **then**
9:             Form sketch $Y \leftarrow G_t \Omega$ with a random Gaussian matrix $\Omega \in \mathbb{R}^{n \times r_{os}}$
10:            Compute thin QR decomposition: $Q_{\text{sketch}}, R_{\text{sketch}} \leftarrow \text{qr}(Y)$
11:            Set projection matrix $Q_t \leftarrow Q_{\text{sketch}}[:, : r]$
12:         **else**
13:            Form sketch $Y \leftarrow G_t^T \Omega$ with a random Gaussian matrix $\Omega \in \mathbb{R}^{m \times r_{os}}$
14:            Compute thin QR decomposition: $Q_{\text{sketch}}, R_{\text{sketch}} \leftarrow \text{qr}(Y)$
15:            Set projection matrix $P_t \leftarrow Q_{\text{sketch}}[:, : r]$
16:         **end if**
17:     **end if**
18:     **if** $m \geq n$ **then**
19:         Project gradient: $G_{\text{proj}} \leftarrow G_t Q_t$
20:         Update low-rank optimizer states: $M_t, V_t \leftarrow \text{AdamW}(G_{\text{proj}}, M_{t-1}, V_{t-1})$
21:         Compute low-rank update $\Delta W_{\text{proj}}$ from $M_t, V_t$
22:         Project back to full space: $\Delta W_t \leftarrow \Delta W_{\text{proj}} Q_t^T$
23:     **else**
24:         Project gradient: $G_{\text{proj}} \leftarrow P_t^T G_t$
25:         Update low-rank optimizer states: $M_t, V_t \leftarrow \text{AdamW}(G_{\text{proj}}, M_{t-1}, V_{t-1})$
26:         Compute low-rank update $\Delta W_{\text{proj}}$ from $M_t, V_t$
27:         Project back to full space: $\Delta W_t \leftarrow P_t \Delta W_{\text{proj}}$
28:     **end if**
29:     Update weights: $W_t \leftarrow W_t - \eta \Delta W_t$
30: **end for**

---

### 3.3 COMPUTATIONAL COMPLEXITY ANALYSIS

We justify GALE's speedup by analyzing its computational cost for subspace identification. Let the gradient matrix be $G \in \mathbb{R}^{m \times n}$ with a target rank $r \ll n$. The original GaLore method relies on a full Singular Value Decomposition (SVD) of $G$, an expensive procedure with $O(mn^2)$ complexity. While recent adaptations like GaLore-2 employ randomized SVD to reduce this cost, they typically rely on iterative solvers. GALE circumvents this bottleneck using a highly efficient, **direct** (non-iterative) two-stage randomized approach. First, it forms a low-dimensional sketch $Y = G\Omega$ by projecting $G$ onto a random matrix $\Omega \in \mathbb{R}^{n \times r_{os}}$, where $r_{os}$ is a small oversampling factor. This dominant step costs only $O(mnr_{os})$. Second, an orthonormal basis is extracted from the much smaller sketch $Y$ via a thin QR decomposition, a fast, **direct** (non-iterative) algorithm costing just $O(mr_{os}^2)$. The total complexity for GALE is thus $O(mnr_{os} + mr_{os}^2)$, a significant improvement over the SVD-based $O(mn^2)$. Since $r_{os} \ll n$, GALE's core achievement is replacing the prohibitively expensive quadratic term $n^2$ with a much more favorable, near-linear term $n \cdot r_{os}$, providing the theoretical foundation for its dramatic empirical speedup.

## 4 EXPERIMENTS

To validate the effectiveness of GALE, we conduct a comprehensive evaluation on two standard and demanding language model training paradigms: pre-training from scratch and downstream fine-tuning. We compare our proposed GALE variants against a full-rank AdamW baseline, the prior state-of-the-art gradient projection method GaLore, and several widely-used PEFT methods (LoRA, $(IA)^3$, and Prefix-Tuning). Our experiments are designed to assess three key dimensions: optimizer

Table 1: Pretraining results for Llama 100M, 250M, and 1B on C4 for 10,000 steps. We compare our GALE methods against full pretraining (AdamW), parameter-efficient fine-tuning (PEFT) baselines applied to pretraining, and GaLore. Memory is reported in MiB, and optimizer update step time in milliseconds (ms).

| | Method | Rank* | Perplexity | Memory (MiB) | | | Step Time (ms) |
|---|---|---|---|---|---|---|---|
| | | | | **Params** | **Opt States** | **Steady State** | |
| Llama 100M | AdamW | – | 29.04 | 190.96 | 381.92 | 641.23 | 2.09 |
| | LoRA | 128 | 8645.21 | 198.46 | 15.00 | 440.60 | 0.71 |
| | IA³ | – | 36 300.61 | 191.00 | 0.09 | 418.25 | 0.84 |
| | Prefix-Tuning | 128 | 11 584.64 | 194.93 | 7.94 | 430.01 | 0.37 |
| | GaLore (AdamW) | 128 | 110.38 | 190.96 | 139.13 | 410.58 | 25.25 |
| | APOLLO | 128 | 108.03 | 190.96 | 139.13 | 410.58 | 10.48 |
| | GALE | 128 | 112.04 | 190.96 | 139.13 | 410.58 | 10.43 |
| | GALE Fused | 128 | 100.03 | 190.96 | 139.13 | 410.58 | 6.15 |
| | GALE Fused Approx | 128 | 102.87 | 190.96 | 139.13 | 410.58 | 5.74 |
| Llama 250M | AdamW | – | 24.80 | 471.82 | 943.64 | 1495.14 | 3.34 |
| | LoRA | 256 | 5634.58 | 508.11 | 72.00 | 850.82 | 1.59 |
| | IA³ | – | 37 276.26 | 471.93 | 0.21 | 740.89 | 1.05 |
| | Prefix-Tuning | 256 | 9154.81 | 490.31 | 36.98 | 796.04 | 0.42 |
| | GaLore (AdamW) | 256 | 60.71 | 471.82 | 377.14 | 991.76 | 55.16 |
| | APOLLO | 256 | 58.11 | 471.82 | 377.14 | 992.14 | 26.08 |
| | GALE | 256 | 58.08 | 471.82 | 377.14 | 992.14 | 25.75 |
| | GALE Fused | 256 | 53.55 | 471.82 | 377.14 | 993.01 | 12.16 |
| | GALE Fused Approx | 256 | 53.48 | 471.82 | 377.14 | 993.01 | 11.13 |
| Llama 1B | AdamW | – | 27.49 | 2554.10 | 5108.20 | 7778.12 | 3.58 |
| | LoRA | 512 | 431.58 | 2746.88 | 384.00 | 3757.69 | 1.04 |
| | IA³ | – | 48 121.93 | 2554.38 | 0.56 | 3182.54 | 0.99 |
| | Prefix-Tuning | 512 | 4369.33 | 2653.18 | 196.61 | 3476.61 | 0.35 |
| | GaLore (AdamW) | 512 | 39.70 | 2554.10 | 1464.84 | 4481.33 | 225.30 |
| | APOLLO | 512 | 39.20 | 2554.10 | 1464.84 | 4509.01 | 19.63 |
| | GALE | 512 | 39.20 | 2554.10 | 1464.84 | 4509.01 | 18.89 |
| | GALE Fused | 512 | 37.54 | 2554.10 | 1464.84 | 4517.00 | 12.09 |
| | GALE Fused Approx | 512 | 37.55 | 2554.10 | 1464.84 | 4514.33 | 9.58 |

*Rank denotes GaLore rank, LoRA rank, or Prefix-Tuning bottleneck size.

update step time (ms), task performance (perplexity and GLUE scores), and memory efficiency. All experiments were conducted on NVIDIA A100 80GB GPUs.

## 4.1 Pretraining Llama Models on C4

We evaluate GALE by pretraining LLaMA models (100M, 250M, 1B) from scratch on the C4 dataset for 10,000 steps. We compare against full-rank AdamW, standard PEFT methods, and gradient projection baselines including GaLore and the recently proposed APOLLO. Our primary memory metric, **Steady State Memory**, measures the total allocated GPU memory after the first optimizer step.

The results confirm that PEFT methods are unsuitable for pretraining, exhibiting high perplexity and potentially larger memory footprints than GALE on smaller models due to architectural overheads. In contrast, gradient projection methods like GALE and APOLLO achieve perplexities comparable to the AdamW baseline by enabling full-parameter updates.

GALE's main contribution is its superior balance of speed and performance. While APOLLO effectively mitigates GaLore's SVD bottleneck—achieving step times comparable to our native implementation (e.g., 19.63ms vs 18.89ms on the 1B model)—GALE's optimized variants deliver strictly

Table 2: Fine-tuning results for BERT-Large and GPT-2-Large on the GLUE benchmark. We compare our GALE methods against full fine-tuning (AdamW), GaLore, and emerging efficient fine-tuning methods (DoRA, VeRA, FourierFT). We report Matthews correlation (MCC) for CoLA, F1 score for MRPC and QQP, Pearson correlation (Pears) for STS-B, and accuracy (Acc) for MNLI, QNLI, RTE, and SST-2

| Method | Rank* | GLUE Task (Metric) | | | | | | | | Memory (MiB) | | | Step Time (ms) |
|---|---|---|---|---|---|---|---|---|---|---|---|---|---|
| | | CoLA (MCC) | MNLI (Acc) | MRPC (F1) | QNLI (Acc) | QQP (F1) | RTE (Acc) | SST-2 (Acc) | STS-B (Pears) | Params | Opt States | Steady State | |
| **BERT-Large** | | | | | | | | | | | | | |
| AdamW | – | 0.581 | 0.867 | 0.872 | 0.901 | 0.825 | 0.704 | 0.929 | 0.876 | 639.24 | 1278.47 | 1935.14 | 3.10 |
| LoRA | 4 | 0.592 | 0.862 | 0.903 | 0.907 | 0.828 | 0.733 | 0.930 | 0.899 | 639.99 | 1.51 | 658.15 | 0.90 |
| IA³ | – | 0.569 | 0.862 | 0.836 | 0.896 | 0.821 | 0.701 | 0.913 | 0.856 | 639.52 | 0.57 | 656.74 | 0.70 |
| Prefix-Tuning | 4 | 0.548 | 0.859 | 0.812 | 0.888 | 0.810 | 0.695 | 0.908 | 0.856 | 639.77 | 1.08 | 657.54 | 0.20 |
| DoRA | 4 | 0.603 | 0.869 | 0.891 | 0.923 | 0.723 | 0.705 | 0.948 | 0.873 | 646.87 | 15.27 | 679.81 | 6.20 |
| VeRA | 4 | 0.615 | 0.868 | 0.891 | 0.921 | 0.720 | 0.702 | 0.949 | 0.871 | 640.09 | 1.71 | 659.80 | 5.30 |
| FourierFT | – | 0.608 | 0.865 | 0.887 | 0.919 | 0.718 | 0.698 | 0.947 | 0.869 | 639.79 | 1.11 | 658.63 | 2.90 |
| GaLore (AdamW) | 4 | 0.579 | 0.865 | 0.812 | 0.901 | 0.818 | 0.701 | 0.913 | 0.864 | 639.24 | 130.97 | 788.77 | 104.40 |
| GALE | 4 | 0.582 | 0.874 | 0.812 | 0.895 | 0.815 | 0.710 | 0.913 | 0.852 | 639.24 | 130.97 | 788.77 | 29.80 |
| GALE Fused | 4 | 0.584 | 0.870 | 0.812 | 0.896 | 0.820 | 0.714 | 0.913 | 0.865 | 639.24 | 130.97 | 788.77 | 14.40 |
| GALE Fused Approx | 4 | 0.578 | 0.870 | 0.812 | 0.895 | 0.824 | 0.704 | 0.913 | 0.852 | 639.24 | 130.97 | 788.77 | 13.90 |
| **GPT-2-Large** | | | | | | | | | | | | | |
| AdamW | – | 0.452 | 0.785 | 0.821 | 0.853 | 0.684 | 0.627 | 0.902 | 0.813 | 1476.35 | 2952.70 | 4558.04 | 3.80 |
| LoRA | 8 | 0.448 | 0.781 | 0.817 | 0.849 | 0.681 | 0.623 | 0.899 | 0.809 | 1479.16 | 5.64 | 1596.13 | 0.90 |
| IA³ | – | 0.445 | 0.778 | 0.814 | 0.846 | 0.679 | 0.620 | 0.897 | 0.806 | 1476.88 | 1.06 | 1606.15 | 0.70 |
| Prefix-Tuning | 8 | 0.432 | 0.769 | 0.801 | 0.835 | 0.672 | 0.611 | 0.891 | 0.798 | 1478.02 | 3.36 | 1599.04 | 0.30 |
| DoRA | 8 | 0.462 | 0.791 | 0.825 | 0.855 | 0.687 | 0.631 | 0.904 | 0.815 | 1482.63 | 6.55 | 345.02 | 2.20 |
| VeRA | 8 | 0.459 | 0.788 | 0.822 | 0.852 | 0.684 | 0.628 | 0.901 | 0.812 | 1477.89 | 0.36 | 333.42 | 1.70 |
| FourierFT | – | 0.455 | 0.784 | 0.819 | 0.849 | 0.681 | 0.625 | 0.898 | 0.809 | 1476.49 | 0.28 | 336.01 | 1.20 |
| GaLore (AdamW) | 8 | 0.450 | 0.783 | 0.815 | 0.851 | 0.683 | 0.625 | 0.898 | 0.811 | 1476.35 | 604.67 | 1824.31 | 20.80 |
| GALE | 8 | 0.451 | 0.784 | 0.816 | 0.852 | 0.683 | 0.626 | 0.898 | 0.811 | 1476.35 | 604.67 | 1824.31 | 10.04 |
| GALE Fused | 8 | 0.453 | 0.786 | 0.818 | 0.854 | 0.685 | 0.628 | 0.900 | 0.813 | 1476.35 | 604.67 | 1824.31 | 5.40 |
| GALE Fused Approx | 8 | 0.452 | 0.785 | 0.817 | 0.853 | 0.685 | 0.627 | 0.900 | 0.813 | 1476.35 | 604.67 | 1824.31 | 3.50 |

*Rank denotes GaLore rank, LoRA rank, or Prefix-Tuning bottleneck size. For DoRA, VeRA, and FourierFT, rank is equivalent effective rank.

better throughput and task performance. **GALE Fused** executes the optimizer step roughly **1.6×** faster than APOLLO (12.09ms vs 19.63ms) on the 1B model while achieving consistently lower perplexity (37.54 vs 39.20). Furthermore, **GALE Fused Approx** is over **23×** faster than GaLore (9.58ms vs 225.30ms) and twice as fast as APOLLO, offering the highest throughput.

In conclusion, GALE outperforms both GaLore and APOLLO, providing a practical solution that delivers the benefits of full-parameter, low-rank gradient training with state-of-the-art computational efficiency.

## 4.2 FINE-TUNING ON GLUE

We evaluate GALE on the GLUE benchmark, a downstream fine-tuning setting where Parameter-Efficient Fine-Tuning (PEFT) methods are known to be exceptionally strong. We expand our evaluation to include not only standard baselines like LoRA but also recent advanced techniques—DoRA, VeRA, and FourierFT—to assess GALE as a general-purpose memory-efficient method against the modern state-of-the-art.

As shown in Table 2, specialized PEFT methods remain a powerful baseline. In particular, DoRA and VeRA demonstrate that parameter-efficient methods can achieve results highly competitive with full fine-tuning. In contrast, the previous gradient projection method, GaLore, exhibits a clear performance deficit compared to these advanced PEFTs.

Our key finding is that GALE significantly reduces the performance gap between gradient projection methods and top-tier PEFTs. GALE consistently improves upon GaLore, establishing a new, much stronger baseline for full-parameter efficient fine-tuning. For instance, on BERT-Large, **GALE Fused** improves the MNLI score from GaLore's 0.865 to 0.870 and the QQP score from 0.818 to 0.820, surpassing even the strong DoRA baseline on MNLI (0.869).

Crucially, GALE achieves this while maintaining high computational efficiency. The **GALE Fused** optimizer step is over **7.2×** faster than GaLore's on BERT-Large (14.40ms vs. 104.40ms) and nearly **4×** faster on GPT-2-Large (5.40ms vs. 20.80ms). This makes full-parameter fine-tuning substantially more practical, narrowing the performance gap with PEFTs without inheriting the computational overhead of SVD-based methods.

Table 3: Pretraining results for Llama 1B on C4 for 10,000 steps, comparing Adafactor and AdamW8bit optimizers with GaLore and GALE methods.

| | **Method** | **Rank** | **Perplexity** | **Memory (MiB)** | | **Step Time (ms)** |
|---|---|---|---|---|---|---|
| | | | | **Params** | **Opt States** | |
| *Llama 1B* | Adafactor | – | 34.81 | 2554.1 | 55.6 | 56.63 |
| | GaLore (Adafactor) | 512 | 25.99 | 2554.1 | 53.8 | 299.47 |
| | GALE (Adafactor) | 512 | 27.61 | 2554.1 | 53.8 | 94.42 |
| | GALE (Adafactor) Fused | 512 | 27.61 | 2554.1 | 53.8 | 93.42 |
| | GALE (Adafactor) Fused Approx | 512 | 27.69 | 2554.1 | 53.8 | 87.49 |
| | AdamW8bit | – | 32.52 | 2554.1 | 2594.33 | 26.77 |
| | GaLore (AdamW8bit) | 512 | 30.22 | 2554.1 | 744.77 | 247.21 |
| | Q-GaLore (AdamW8bit) | 512 | 31.06 | 2554.1 | 524.12 | 265.34 |
| | GALE (AdamW8bit) | 512 | 29.87 | 2554.1 | 744.77 | 39.34 |
| | GALE (AdamW8bit) Fused | 512 | 29.87 | 2554.1 | 744.77 | 39.38 |
| | GALE (AdamW8bit) Fused Approx | 512 | 29.86 | 2554.1 | 744.77 | 36.84 |

While PEFTs remain an excellent choice for fine-tuning-only scenarios, GALE's ability to outperform strong baselines like DoRA in specific tasks, combined with its state-of-the-art performance in pre-training, positions it as a more versatile solution.

### 4.3 Synergy with Memory-Efficient Optimizers

GALE's focus on gradient projection makes it orthogonal to methods that modify the optimizer states directly. To test this, we combine GALE with two popular memory-efficient optimizers, Adafactor and 8-bit AdamW, on the 1B Llama pre-training task. We also benchmark against **Q-GaLore**, which integrates quantization directly into the projection steps. The results (Table 3) confirm that GALE acts as a powerful synergistic component.

With Adafactor, GALE significantly lowers perplexity compared to the baseline (27.61 vs. 34.81) while being over **3.1× faster** than GaLore-Adafactor. The synergy with 8-bit AdamW is even more compelling. While Q-GaLore achieves the lowest memory footprint (524.12 MiB) via aggressive INT4 quantization, it suffers from high computational overhead (265.34ms) and degraded perplexity (31.06). In contrast, GALE (AdamW8bit) Fused is over **6.7× faster** than Q-GaLore (39.38ms), achieves significantly better perplexity (29.87), and maintains a competitive memory footprint. These experiments demonstrate that GALE is a complementary module that effectively balances memory, speed, and final task performance better than aggressive quantization alternatives.

### 4.4 Ablation Study : Projection Rank and Refresh Gap

To understand the trade-offs between computational cost and model performance, we ablate the two primary hyperparameters for gradient projection: the **projection rank** ($r$) and the **refresh gap** ($k_{\mathbf{upd}}$). Pre-training a 100M LLaMA model on C4 (Figure 2) reveals that GALE achieves comparable or superior perplexity to GaLore across nearly all configurations. Interestingly, when refresh frequency is high (precisely when We hypothesize that GALE's randomized sketching acts as an implicit regularizer, identifying a more robust subspace than an SVD potentially overfit to a single noisy mini-batch gradient.

More critically, the analysis reveals a stark difference in the speed-performance **Pareto frontier**. GALE operates in the highly desirable regime of low perplexity and fast, stable update step times (5–15ms). In contrast, GaLore requires much slower and more variable times (10–80ms) to achieve similar performance. This is a direct consequence of their computational complexities; GaLore's throughput is highly sensitive to the refresh gap, as its runtime is dominated by the expensive SVD ($O(mn^2)$). GALE's asymptotically cheaper randomized QR approach ($O(mnr_{os})$) is not a bottleneck, making its throughput far less sensitive to how often the basis is refreshed. GALE thus

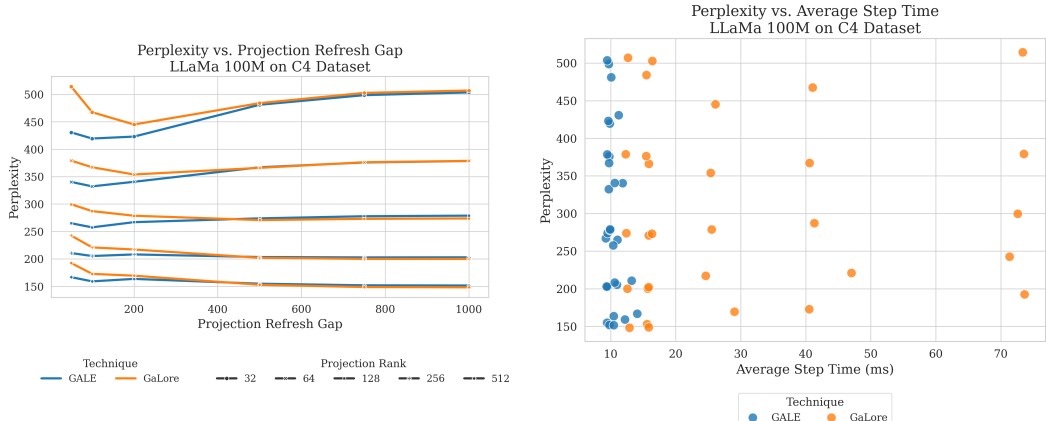

Figure 2: **(Left)** Ablation study of perplexity vs projection refresh gap for different projection ranks. The model is a 100M Llama model, pretrained on C4 for 10,000 steps. **(Right)** A scatter plot of perplexity vs average optimizer update step time for different projection refresh gaps (200,400,600,800,1000) and projection ranks (32,64,128,256,512)

establishes a superior Pareto frontier and offers a more stable, predictable training time, moderately decoupling throughput from hyperparameter tuning.

## 4.5 ABLATION STUDY: GALE HYPERPARAMETERS AND THEIR SYNERGY

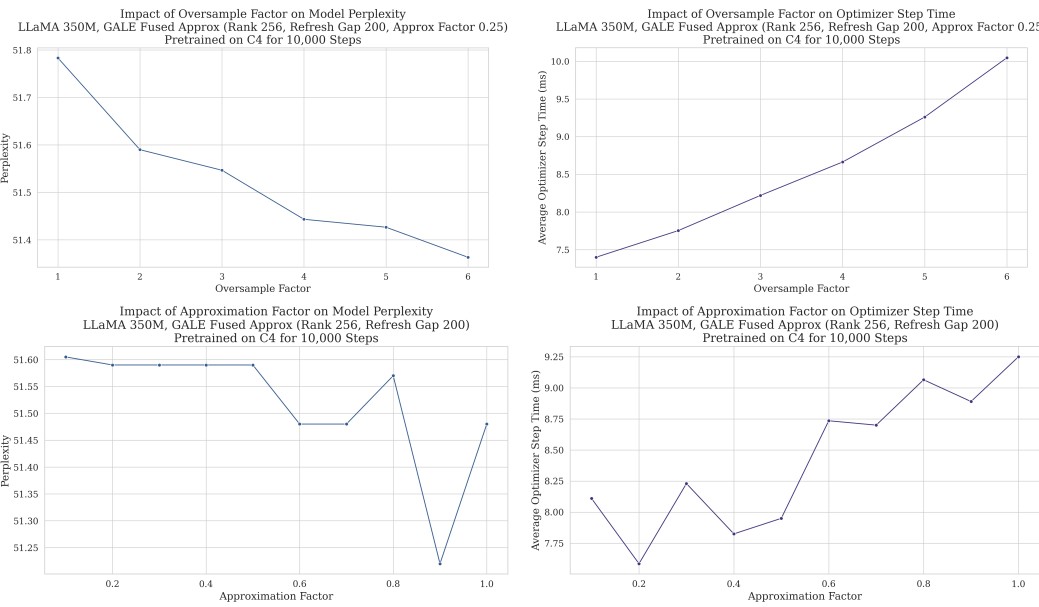

Figure 3: Ablation study on GALE's key hyperparameters. **(Top Row)** The effect of the oversampling factor ($f_{os}$) on final perplexity and optimizer step time. Increasing $f_{os}$ modestly improves perplexity at a linear cost to speed. **(Bottom Row)** The effect of the approximation factor ($f_{\text{approx}}$) on perplexity and step time. Task performance is remarkably robust to aggressive subsampling, making it a powerful lever for speed. Experiments were run on a 350M Llama model with $r = 256$ and $k_{\text{upd}} = 200$.

To guide hyperparameter tuning, we analyze GALE's two main performance-related hyperparameters: the **oversampling factor** ($f_{os}$) and the **approximation factor** ($f_{\text{approx}}$). The former improves the quality of the initial random sketch, while the latter (unique to GALE Fused Approx) accelerates

Table 4: Multi-Scale Parallelism Performance (rank=128)

| | Configuration | Memory (MiB) | | | Throughput (samp/s) |
| | | Params | Optimizer State | Steady State | |
|---|---|---|---|---|---|
| 60M | 1-GPU + GALE | 110.8 | 52.0 | 187.1 | 7.72 |
| | 2-GPU DDP + GALE | 146.4 | 74.7 | 455.8 | 22.33 |
| | 2-GPU FSDP + GALE | **110.8** | **56.4** | **337.9** | 49.46 |
| 100M | 1-GPU + GALE | 191.0 | 70.8 | 292.1 | 5.56 |
| | 2-GPU DDP + GALE | 235.3 | 97.8 | 670.5 | 15.30 |
| | 2-GPU FSDP + GALE | **191.0** | **97.1** | **548.4** | 34.48 |
| 350M | 1-GPU + GALE | 701.9 | 145.1 | 926.3 | 3.13 |
| | 2-GPU DDP + GALE | 773.2 | 185.8 | 1928.2 | 8.70 |
| | 2-GPU FSDP + GALE | **701.9** | **356.8** | **1869.9** | 18.71 |
| 1B | 1-GPU + GALE | 2554.1 | 281.7 | 2979.5 | 2.22 |
| | 2-GPU DDP + GALE | 2696.7 | 358.6 | 6054.9 | 5.11 |
| | 2-GPU FSDP + GALE | **2554.1** | **1297.6** | **6566.0** | 11.72 |

the final orthonormalization step by subsampling the sketch matrix. We pre-train a 350M LLaMA model on C4 to analyze their effects, shown in Figure 3.

The results reveal key trade-offs and a powerful synergy. Increasing $f_{os}$ modestly lowers perplexity at a near-linear cost to the optimizer step time. Conversely, task performance is remarkably robust to $f_{approx}$, allowing for aggressive subsampling to significantly reduce latency with minimal impact on perplexity. The most effective strategy is to combine these effects: invest computation in a high-quality initial sketch (a larger $f_{os}$) and then recover that cost with aggressive approximation (a smaller $f_{approx}$). This synergistic approach can yield both better performance and faster step times than a minimal, non-approximated sketch. This synergy provides flexible control over the speed-performance trade-off. For general use, we find defaults of $f_{os} = 2$ and $f_{approx} = 0.2$ offer an excellent balance.

## 4.6 MULTI-GPU SCALING EFFICIENCY

We evaluate GALE's multi-GPU scalability using Distributed Data Parallel (DDP) and Fully Sharded Data Parallel (FSDP) across four model sizes (60M, 100M, 350M, 1B) on WikiText-2 with rank=128.

Table 4 shows key findings. First, GALE's optimizer memory scales sub-linearly: optimizer-to-parameter ratio decreases from 0.47× (60M) to 0.11× (1B), as fixed-rank projection becomes more efficient for larger models. Second, FSDP maintains strong throughput with 597% average improvement over single-GPU (range: 529–641%), approaching theoretical 2× scaling with doubled batch size. The 1B model achieves 11.72 vs 2.22 samples/s, confirming GALE's infrequent projection updates (every 50 steps) impose minimal synchronization cost.

These results establish GALE scales effectively to multi-GPU configurations, with FSDP providing optimal memory-throughput balance for memory-constrained scenarios.

## 5 CONCLUSION

Existing efficient LLM training methods present a difficult compromise: PEFTs trade performance for speed, while gradient projection methods like GaLore trade speed for memory. GaLore's reliance on Singular Value Decomposition (SVD) creates a computational bottleneck, undermining its practicality. We introduced **GALE**, which resolves this trade-off by replacing the expensive SVD with a fast, non-iterative randomized QR decomposition. This minimizes GaLore's optimizer overhead and outperforms recent approximations, delivering memory savings with superior computational speed. Our experiments confirm GALE's advantages. In pre-training, GALE matches GaLore's task performance while executing optimizer steps up to 23x faster. In fine-tuning, GALE narrows the performance gap relative to GaLore when compared with specialized PEFTs, making full-parameter

training a more viable alternative. By decoupling memory savings from computational cost, GALE makes full-parameter LLM training more practical and lowers the hardware barrier for developing large-scale models.

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

## ACKNOWLEDGMENTS

We acknowledge the use of AI tools to improve the writing clarity and compress content to fit the page limit. The AI-generated content was carefully reviewed and edited by the authors, who take full responsibility for the final manuscript content.

# A    THEORETICAL ANALYSIS

This appendix provides the detailed theoretical underpinnings for GALE, as referenced in the main text. We first establish a formal bound on the gradient approximation error from our randomized projection. We then provide a convergence guarantee for GALE and finish with a computational complexity analysis.

## A.1    BOUND ON THE GRADIENT PROJECTION ERROR

The efficacy of GALE hinges on its ability to identify a high-quality low-rank subspace without resorting to a costly SVD. The following theorem, based on foundational work in randomized numerical linear algebra (Halko et al., 2011), proves that the subspace identified by our sketching and QR decomposition procedure is a strong approximation of the optimal subspace defined by the top singular vectors, with a provably small expected error.

**Theorem 1** (Randomized Subspace Approximation Error). *Let $G \in \mathbb{R}^{m \times n}$ be the gradient matrix, $r$ be the target rank, and $r_{os}$ be the oversampled rank used for sketching, such that $r_{os} \geq r + 2$. Without loss of generality, assume $m \geq n$. Let $Q \in \mathbb{R}^{n \times r}$ be the orthonormal basis produced by the randomized sketching and QR procedure in Algorithm 1. Let $P_Q = QQ^T$ be the orthogonal projection operator onto the subspace spanned by the columns of $Q$. Let $\sigma_j(G)$ be the $j$-th largest singular value of $G$. The expected approximation error is bounded by:*

$$\mathbb{E}\left[\|G - GP_Q\|_F^2\right] \leq \left(1 + \frac{r}{r_{os} - r - 1}\right) \sum_{j=r+1}^{n} \sigma_j(G)^2 \qquad (2)$$

*An analogous bound holds for the row space projection when $m < n$ by considering $G^T$.*

*Proof Sketch.* The full proof is detailed in (Halko et al., 2011). The core insight is that randomized sketching (e.g., forming $Y = G\Omega$) acts as a low-dimensional embedding that preserves the dominant spectral properties of $G$. The QR decomposition then extracts a stable orthonormal basis for this captured subspace. The term $\sum_{j=r+1}^{n} \sigma_j(G)^2$ represents the error of the *optimal* rank-$r$ approximation given by a truncated SVD. Our randomized method's expected error is guaranteed to be within a small factor, $\left(1 + \frac{r}{r_{os}-r-1}\right)$, of this optimal error. With modest oversampling (e.g., $r_{os} = 2r$), this factor is close to 1, ensuring a high-quality approximation. □

This theorem provides a formal upper bound on the information loss, measured in terms of Frobenius norm, incurred by our randomized projection. The bound guarantees a small error if the gradient's singular values exhibit rapid decay, a property that is empirically validated for large neural networks.

## A.2 CONVERGENCE GUARANTEE FOR GALE-BASED TRAINING

To provide theoretical intuition on the impact of our gradient projection, we analyze the convergence of an idealized GALE-based optimizer in a simplified setting. We consider full-batch projected gradient descent where the projection basis is recomputed at each step. While our practical algorithm employs a stochastic, adaptive optimizer (AdamW) and periodic basis updates for efficiency, this analysis isolates the effect of the projection error and demonstrates that convergence is fundamentally tied to the quality of the approximation established in Theorem 1.

**Theorem 2** (Convergence of Projected Gradient Descent). *Let the loss function $\mathcal{L}(W)$ be L-smooth and be bounded below by $\mathcal{L}^*$. Consider the projected gradient descent algorithm with the update rule $W_{t+1} = W_t - \eta \Delta W_t$, where the update $\Delta W_t = G_t P_t$ is the projection of the true gradient $G_t = \nabla_{W_t} \mathcal{L}$ onto the subspace defined by the GALE projection operator $P_t = Q_t Q_t^T$. With a learning rate $\eta \leq 1/L$, the algorithm converges to a stationary point in expectation:*

$$\frac{1}{T} \sum_{t=0}^{T-1} \mathbb{E}\left[\|G_t\|_F^2\right] \leq \frac{\mathcal{L}(W_0) - \mathcal{L}^*}{T\eta} + \frac{1}{T} \sum_{t=0}^{T-1} \mathbb{E}\left[\epsilon_t^2\right] \tag{3}$$

*where $\epsilon_t^2 = \|G_t - G_t P_t\|_F^2$ is the projection error at step $t$.*

*Proof.* By the $L$-smoothness of $\mathcal{L}$, we have the standard descent lemma:

$$\mathcal{L}(W_{t+1}) \leq \mathcal{L}(W_t) + \langle G_t, W_{t+1} - W_t \rangle + \frac{L}{2}\|W_{t+1} - W_t\|_F^2 \tag{4}$$

$$= \mathcal{L}(W_t) - \eta\langle G_t, G_t P_t \rangle + \frac{L\eta^2}{2}\|G_t P_t\|_F^2 \tag{5}$$

Since $P_t$ is an orthogonal projection, $\langle G_t, G_t P_t \rangle = \|G_t P_t\|_F^2$. For a learning rate $\eta \leq 1/L$, this simplifies to $\mathcal{L}(W_{t+1}) \leq \mathcal{L}(W_t) - \frac{\eta}{2}\|G_t P_t\|_F^2$. By the Pythagorean theorem for projections, $\|G_t P_t\|_F^2 = \|G_t\|_F^2 - \|G_t - G_t P_t\|_F^2 = \|G_t\|_F^2 - \epsilon_t^2$. Substituting this back:

$$\mathcal{L}(W_{t+1}) \leq \mathcal{L}(W_t) - \frac{\eta}{2}(\|G_t\|_F^2 - \epsilon_t^2) \tag{6}$$

Rearranging gives $\|G_t\|_F^2 \leq \frac{2}{\eta}(\mathcal{L}(W_t) - \mathcal{L}(W_{t+1})) + \epsilon_t^2$. Taking the expectation and summing from $t = 0$ to $T - 1$ yields a telescoping sum. Since $\mathcal{L}(W_T) \geq \mathcal{L}^*$, dividing by $T$ yields the final result. As $T \to \infty$, the first term vanishes, and convergence is determined by the average projection error. As Theorem 1 guarantees $\mathbb{E}[\epsilon_t^2]$ is small and bounded, the average gradient norm is driven towards a small value. □

**Remark on Periodic Updates.** Our proof assumes the projection $P_t$ is optimal for the current gradient $G_t$. In practice, we update the basis every $k_{\text{upd}}$ steps, using a stale projection for intermediate steps. This introduces an additional error term related to how quickly the gradient subspace changes. The empirical success of our method suggests that this subspace evolves slowly enough that periodic updates are sufficient to maintain effective convergence.

## A.3 COMPUTATIONAL COMPLEXITY ANALYSIS

We justify GALE's speedup by analyzing its computational cost for subspace identification. Let the gradient matrix be $G \in \mathbb{R}^{m \times n}$ with a target rank $r \ll n$. Prior methods rely on a full Singular Value Decomposition (SVD) of $G$, an expensive, **iterative** procedure with $O(mn^2)$ complexity. GALE circumvents this bottleneck using a highly efficient, two-stage randomized approach. First, it forms a low-dimensional sketch $Y = G\Omega$ by projecting $G$ onto a random matrix $\Omega \in \mathbb{R}^{n \times r_{os}}$, where $r_{os}$ is a small oversampling factor. This dominant step costs only $O(mnr_{os})$. Second, an orthonormal basis is extracted from the much smaller sketch $Y$ via a thin QR decomposition, a fast, **direct** (non-iterative) algorithm costing just $O(mr_{os}^2)$. The total complexity for GALE is thus $O(mnr_{os} + mr_{os}^2)$, a significant improvement over the SVD-based $O(mn^2)$. Since $r_{os} \ll n$, GALE's core achievement is replacing the prohibitively expensive quadratic term $n^2$ with a much more favorable, near-linear term $n \cdot r_{os}$, providing the theoretical foundation for its dramatic empirical speedup.

## A.4 Implementation Details for GALE Variants

### A.4.1 GALE Fused: Mixed-Precision Fused Kernels

The **GALE Fused** variant accelerates the optimizer step by replacing standard PyTorch operations with a custom CUDA kernel. This kernel fuses the entire AdamW update for the low-rank projected tensors into a single GPU operation. The benefits are twofold:

1. **Reduced Kernel Launch Overhead:** A single kernel launch has significantly less overhead than launching multiple kernels for element-wise addition, multiplication, division, and square root operations as required by a standard AdamW implementation.

2. **Improved Memory Bandwidth Utilization:** The kernel is designed to maximize data reuse within the GPU's fast shared memory and registers. Low-rank optimizer states $(M_t, V_t)$ and the projected gradient $(G_{\text{proj}})$ are loaded from slower global memory once, the full update is computed, and the new states are written back once.

Furthermore, the kernel employs a mixed-precision strategy to balance speed and numerical stability. The optimizer states are stored in memory-efficient 'bfloat16' format. During the computation, they are cast to 'float32' to perform all arithmetic updates with full precision, preventing the loss of accuracy that can occur with low-precision accumulators. The final updated states are then cast back to 'bfloat16' for storage. This approach leverages the high throughput of 'bfloat16' for memory access while retaining the numerical stability of 'float32' for the sensitive update calculations.

### A.4.2 GALE Fused Approx: Subsampled Sketching

The **GALE Fused Approx** variant pushes throughput further by accelerating the QR decomposition stage itself. The standard randomized algorithm first creates an oversampled sketch matrix $Y \in \mathbb{R}^{d \times r_{os}}$. Instead of performing a QR decomposition on this entire matrix, GALE Fused Approx first creates an approximate sketch $Y_{\text{approx}}$ by randomly selecting a subset of $c$ columns from $Y$, where $c = \max(r, \lfloor r_{os} \cdot f_{\text{approx}} \rfloor)$.

The QR decomposition is then performed on the much smaller matrix $Y_{\text{approx}} \in \mathbb{R}^{d \times c}$. The complexity of this step is reduced from $O(dr_{os}^2)$ to $O(dc^2)$. The hyperparameter $f_{\text{approx}} \in (0, 1]$ provides a direct control knob for the trade-off:

- As $f_{\text{approx}} \to 1$, $c \to r_{os}$, and the method becomes identical to the standard GALE projection, with maximum accuracy but higher latency.

- As $f_{\text{approx}} \to 0$, $c$ becomes smaller (approaching the minimum rank $r$), making the QR decomposition faster but potentially increasing the projection error from Theorem 1, as the basis is derived from less information.

This variant is particularly useful in scenarios where maximum training speed is paramount, and a slight, controlled increase in the gradient approximation error is acceptable.

## A.5 Contextualizing Optimizer Speedup and Overall Throughput

While GALE achieves a substantial speedup of up to $23\times$ in the optimizer update step time compared to GaLore, it is crucial to contextualize this improvement in terms of end-to-end training throughput. A full training iteration consists of multiple stages: data loading, the forward pass, the backward pass, and the optimizer step. For large models, the forward and backward passes constitute the vast majority of the wall-clock time per step. GALE's innovations are specifically targeted at accelerating the optimizer step, which is the primary bottleneck introduced by prior gradient projection methods.

The results in Table 5 reflect this reality. We observe that GaLore incurs a throughput penalty, processing fewer tokens per second (10103) than even the full-rank AdamW baseline (10346). This demonstrates that the overhead of its SVD-based optimizer step is significant enough to slow down the entire training loop. In contrast, GALE variants successfully remove this bottleneck, restoring throughput to a level comparable to the AdamW baseline (e.g., 10233 for GALE Fused Approx).

Table 5: Pretraining results for Llama 1B on C4 for 10,000 steps, showing average training throughput. Memory is reported in MiB, and throughput in tokens/sec.

| Method | Rank* | Perplexity | Params | Opt States | Steady State | Avg. Throughput (tokens/sec) |
|---|---|---|---|---|---|---|
| AdamW | – | 27.49 | 2554.10 | 5108.20 | 7778.12 | 10 346 |
| LoRA | 512 | 431.58 | 2746.88 | 384.00 | 3757.69 | 12 379 |
| IA³ | – | 48 121.93 | 2554.38 | 0.56 | 3182.54 | 11 882 |
| Prefix-Tuning | 512 | 4369.33 | 2653.18 | 196.61 | 3476.61 | 12 277 |
| GaLore (AdamW) | 512 | 39.70 | 2554.10 | 1464.84 | 4481.33 | 10 103 |
| GALE | 512 | 39.20 | 2554.10 | 1464.84 | 4509.01 | 10 202 |
| GALE Fused | 512 | 37.54 | 2554.10 | 1464.84 | 4517.00 | 10 216 |
| GALE Fused Approx | 512 | 37.55 | 2554.10 | 1464.84 | 4514.33 | 10 233 |

*Rank denotes GaLore rank, LoRA rank, or Prefix-Tuning bottleneck size.

Table 6: Pretraining results for Llama 100M on C4 for 10,000 steps. DoRA, VeRA, and FourierFT show similar catastrophic underfitting as other low-rank methods when used for pretraining from scratch.

| Method | Rank/Budget* | Perplexity | Params | Opt States | Steady State | Step Time (ms) |
|---|---|---|---|---|---|---|
| AdamW | – | 29.04 | 190.96 | 381.92 | 641.23 | 2.09 |
| LoRA | 128 | 8600.9 | 198.46 | 15.00 | 440.60 | 0.71 |
| IA³ | – | 35 010.6 | 191.00 | 0.09 | 418.25 | 0.84 |
| Prefix-Tuning | 128 | 11 220.4 | 194.93 | 7.94 | 430.01 | 0.37 |
| DoRA | 128 | 8702.2 | 262.53 | 143.15 | 423.79 | 8.40 |
| VeRA | 128 | 8206.1 | 191.30 | 0.69 | 211.76 | 2.54 |
| FourierFT | 1000 | 8450.3 | 191.28 | 0.64 | 210.02 | 2.12 |
| GaLore (AdamW) | 128 | 110.9 | 190.96 | 139.13 | 410.58 | 25.25 |
| GALE | 128 | 114.5 | 190.96 | 139.13 | 410.58 | 10.43 |
| GALE Fused | 128 | 102.3 | 190.96 | 139.13 | 410.58 | 6.15 |
| GALE Fused Approx | 128 | 101.9 | 190.96 | 139.13 | 410.58 | 5.74 |

*Rank denotes GaLore rank, LoRA rank, or Prefix-Tuning bottleneck size. For FourierFT, spectral coefficient count is shown.

Therefore, the key contribution of GALE is not a dramatic acceleration of the entire training process, but rather the elimination of the computational overhead inherent in previous gradient projection techniques. By making the optimizer step for memory-efficient, full-parameter training computationally inexpensive, GALE makes this class of methods practically viable, delivering significant memory savings without sacrificing the training throughput that was previously lost.

## A.6 ADDITIONAL PEFT BENCHMARKS ON PRETRAINING FROM SCRATCH

Table 1 made it clear accross several model sizes that PEFT methods do not perform well on pretraining from scratch for LoRA, $IA^3$ and Prefix-Tuning. It is reasonable to believe that this extends to other PEFT methods such as VeRA, DoRA and FourierFT at face value. Nonetheless, we provide a benchmark for them on one of the Llama models (100M) in Table 6 in order to validate this assumption and also inspect the memory saving benefits of these SOTA PEFT techniques on Llama models.

## A.7 APPENDIX: LLAMA 7B SCALABILITY VALIDATION

To validate GALE's applicability to production-scale models, we conducted experiments on LLaMA 7B (6.74B parameters) using a single NVIDIA A40 GPU (48GB VRAM). Table 7 presents memory and performance metrics demonstrating that GALE enables billion-parameter model training on consumer/research-grade hardware.

Table 7: LLaMA 7B memory and performance on single 48GB GPU. All configurations use GALE+AdamW8bit with BFloat16 precision, activation checkpointing, and per-layer optimization.

| Config | Rank | Seq | Param (GB) | Opt (MB) | Steady State (GB) | Throughput (tok/s) |
|---|---|---|---|---|---|---|
| GALE r=128 | 128 | 512 | 12.55 | 371.82 | 38.47 | 142.7 |
| GALE r=256 | 256 | 512 | 12.55 | 527.64 | 38.63 | 143.1 |
| GALE r=128 | 128 | 768 | 12.55 | 371.82 | 39.14 | 99.8 |

**Key findings**: GALE achieves $36\times$ optimizer memory compression (371.82 MB vs 13.48 GB for standard AdamW8bit), enabling steady state memory of 38.47 GB on a single 48GB GPU with 20% headroom. Standard AdamW8bit without GALE would exceed the 48GB memory budget, making GALE essential for single-GPU 7B training. The optimizer-to-parameter memory ratio of $0.029\times$ (vs $0.29\times$ at 1B scale with r=512) demonstrates sub-linear scaling, where GALE becomes increasingly efficient for larger models.

These results validate GALE's scalability to production-scale models. Combined with activation checkpointing and per-layer optimization, GALE enables 7B training on hardware previously requiring 80GB A100 GPUs, expanding access to large-scale model development for resource-constrained researchers.

## A.8 EXTENDED PRETRAINING RESULTS

Table 8: Pretraining results for Llama 1B on C4 for 100,000 steps, comparing AdamW8bit optimizer with GaLore and GALE methods. Memory is reported in MiB, and optimizer update step time in milliseconds (ms).

| | Method | Rank | Perplexity | Memory (MiB) | | Step Time (ms) |
|---|---|---|---|---|---|---|
| | | | | **Params** | **Opt States** | |
| *Llama 1B* | GaLore (AdamW8bit) | 512 | 15.57 | 2554.1 | 744.77 | 256.13 |
| | GALE (AdamW8bit) Fused Approx | 512 | 15.38 | 2554.1 | 744.77 | 41.84 |

## A.9 INFORMATION RETENTION: ONE-SIDED QR PROJECTION VS TWO-SIDED SVD

A natural theoretical question arises: how does GALE's one-sided QR projection compare to the optimal two-sided SVD projection in terms of information retention? We provide a rigorous analysis addressing both the structural question (is one-sided projection inherently lossy?) and the algorithmic question (how much error does randomized approximation introduce?).

### A.9.1 SETUP AND NOTATION

Let $G \in \mathbb{R}^{m \times n}$ be the gradient matrix with compact SVD $G = U\Sigma V^T = \sum_{i=1}^{k} \sigma_i u_i v_i^T$, where $k = \min(m, n)$ and $\sigma_1 \geq \sigma_2 \geq \cdots \geq \sigma_k \geq 0$. We partition at rank $r < k$:

- $U_r = [u_1, \ldots, u_r] \in \mathbb{R}^{m \times r}$: top-$r$ left singular vectors

- $V_r = [v_1, \ldots, v_r] \in \mathbb{R}^{n \times r}$: top-$r$ right singular vectors

- $\Sigma_r = \text{diag}(\sigma_1, \ldots, \sigma_r)$: top-$r$ singular values

The **tail energy** $\tau_r^2 = \sum_{i=r+1}^{k} \sigma_i^2$ represents the energy in discarded components. By the Eckart-Young-Mirsky theorem, the truncated SVD $G_r^{\text{SVD}} = U_r \Sigma_r V_r^T$ achieves minimum error: $\|G - G_r^{\text{SVD}}\|_F^2 = \tau_r^2$.

### A.9.2 STRUCTURAL RESULT: ONE-SIDED PROJECTION IS OPTIMAL

The two-sided SVD projection can be written as $G_r^{\text{SVD}} = U_r U_r^T G V_r V_r^T$, projecting onto both the column space (via $U_r U_r^T$) and row space (via $V_r V_r^T$). In contrast, GALE uses a one-sided column space projection: $G_r^Q = QQ^T G$.

**Theorem 3** (Equivalence of Optimal One-Sided and Two-Sided Projection). *The optimal one-sided column space projection achieves the same error as the truncated SVD:*

$$\min_{Q \in \mathbb{R}^{m \times r}, Q^T Q = I_r} \|G - QQ^T G\|_F^2 = \tau_r^2 \tag{7}$$

*The minimum is achieved when $Q = U_r$, and in this case $U_r U_r^T G = G_r^{SVD}$.*

*Proof.* For any orthonormal $Q$, the projection error satisfies $\|G - QQ^T G\|_F^2 = \|G\|_F^2 - \|Q^T G\|_F^2$ by the Pythagorean theorem. Thus minimizing error is equivalent to maximizing $\|Q^T G\|_F^2$.

Using $G = U\Sigma V^T$ and noting that range$(G) \subseteq$ range$(U)$, we can restrict to $Q$ with range$(Q) \subseteq$ range$(U)$ without loss of generality. Let $A = U^T Q \in \mathbb{R}^{k \times r}$; under this restriction, $A^T A = I_r$. Then:

$$\|Q^T G\|_F^2 = \text{tr}(\Sigma^2 U^T QQ^T U) = \text{tr}(A^T \Sigma^2 A) \tag{8}$$

Since $\Sigma^2 = \text{diag}(\sigma_1^2, \ldots, \sigma_k^2)$ is diagonal with decreasing entries, the maximum over orthonormal $A$ is $\sum_{i=1}^r \sigma_i^2$, achieved when $A = [e_1, \ldots, e_r]$ (first $r$ standard basis vectors), corresponding to $Q = U_r$.

For verification: $U_r U_r^T G = U_r U_r^T (U_r \Sigma_r V_r^T + U_\perp \Sigma_\perp V_\perp^T) = U_r \Sigma_r V_r^T = G_r^{\text{SVD}}$, using orthogonality $U_r^T U_\perp = 0$. $\qquad\square$

[Row Space Projection is Redundant] The two-sided projection $U_r U_r^T G V_r V_r^T$ equals the one-sided projection $U_r U_r^T G$.

*Proof.* From Theorem 3, $U_r U_r^T G = U_r \Sigma_r V_r^T$. The row space of this matrix is span$(V_r)$, so right-multiplication by $V_r V_r^T$ is the identity: $(U_r \Sigma_r V_r^T) V_r V_r^T = U_r \Sigma_r V_r^T$. $\qquad\square$

**Interpretation:** There is no inherent information loss from one-sided projection. The row space projection in SVD is mathematically redundant when the column space is correctly identified. This resolves the structural question: one-sided projection onto $U_r$ achieves exactly optimal error.

### A.9.3 ALGORITHMIC RESULT: GALE'S RANDOMIZED APPROXIMATION

GALE does not compute $U_r$ exactly. It approximates the column space via randomized sketching: draw random $\Omega \in \mathbb{R}^{n \times \ell}$ with $\ell = r + p$, compute $Y = G\Omega$, orthonormalize to get $Q$, and project $\tilde{G} = QQ^T G$.

**Theorem 4** (Expected Error Bound, (Halko et al., 2011), Corollary 10.9). *For Gaussian $\Omega$ with $\ell = r + p$ columns where $p \geq 2$:*

$$\mathbb{E}[\|G - QQ^T G\|_F^2] \leq \left(1 + \frac{r}{p-1}\right) \tau_r^2 \tag{9}$$

For GALE's default setting $\ell = 2r$ (so $p = r$), the multiplicative factor is $1 + r/(r-1) = (2r - 1)/(r-1) \approx 2$ for moderate $r$. This worst-case bound is loose; tighter bounds exploit spectral decay.

**Theorem 5** (Error with Spectral Decay, (Halko et al., 2011), Theorem 10.6). *For Gaussian $\Omega$ with $\ell = r + p$ where $p \geq 2$:*

$$\mathbb{E}[\|G - QQ^T G\|_F] \leq \left(1 + \sqrt{\frac{r}{p-1}}\right) \tau_r + \frac{e\sqrt{r+p}}{p} \sigma_{r+1} \tag{10}$$

The second term involves $\sigma_{r+1}$. For matrices with spectral decay ($\sigma_{r+1} \ll \tau_r$), this term is small and the error approaches optimal.

### A.9.4 QUANTITATIVE EXAMPLE

Consider a gradient matrix with $\|G\|_F^2 = 1$, where top-$r$ components capture 95% of energy ($\tau_r^2 = 0.05$), with $r = p = 128$.

**SVD information retention:** $\eta_{\text{SVD}} = 95\%$

**GALE worst-case retention:** $\mathbb{E}[\eta_{\text{GALE}}] \geq 1 - (1 + 128/127) \times 0.05 \approx 90\%$

The gap is at most 5 percentage points, or approximately 5.3% relative to what SVD captures.

### A.9.5 DESCENT DIRECTION PRESERVATION

**Theorem 6** (Descent Preservation). *For any orthonormal $Q$ with $Q^T G \neq 0$:*

$$\langle G, QQ^T G \rangle = \|Q^T G\|_F^2 > 0 \tag{11}$$

*Thus $-QQ^T G$ is a valid descent direction.*

Even imperfect subspace approximation produces valid descent directions, ensuring optimization proceeds correctly.

### A.9.6 SUMMARY

Table 9: Information retention and computational cost comparison.

| Method | Error (Frobenius$^2$) | Complexity |
|---|---|---|
| Truncated SVD | $\tau_r^2$ (optimal) | $O(mnk)$ |
| Optimal one-sided ($Q = U_r$) | $\tau_r^2$ (same) | $O(mnk)$ |
| GALE (worst-case) | $\leq (1 + \frac{r}{p-1})\tau_r^2$ | $O(mnr)$ |
| GALE (with spectral decay) | $\approx \tau_r^2$ | $O(mnr)$ |

The information retention trade-off has two components:

1. **Structural (exact):** One-sided projection with optimal $Q = U_r$ achieves the same error as two-sided SVD. The row space projection is redundant. There is no inherent information loss.

2. **Algorithmic (bounded):** GALE's randomized approximation introduces worst-case error at most $2\times$ optimal for $p = r$. Under spectral decay (typical for neural network gradients), error approaches optimal.

We acknowledge that worst-case bounds are loose without spectral decay assumptions. However, the theoretical framework correctly predicts that performance depends on spectral structure, and empirical studies consistently show neural network gradients exhibit rapid spectral decay, placing them in the favorable regime. This is characteristic of randomized numerical linear algebra: worst-case bounds are pessimistic, but algorithms perform well because real matrices have favorable structure.

### A.10 EMPIRICAL VALIDATION OF GRADIENT SPECTRAL DECAY

The theoretical analysis in Section A.9 establishes that GALE's approximation error depends on the spectral structure of the gradient matrix—specifically, that error approaches optimal when gradients exhibit rapid singular value decay. We provide empirical validation of this assumption by analyzing gradient spectra during LLaMA pretraining.

### A.10.1 EXPERIMENTAL SETUP

We instrument training runs to capture full gradient matrices for all linear layers at steps $t \in \{50, 100, 200, 500\}$. For each captured gradient $G \in \mathbb{R}^{m \times n}$, we compute the full SVD and measure:

- **Power law exponent $\alpha$:** Estimated via linear regression of $\log(\sigma_i)$ vs $\log(i)$, where $\sigma_i \propto i^{-\alpha}$

- **Effective rank**: $r_{\text{eff}} = (\sum_i \sigma_i)^2 / \sum_i \sigma_i^2$

- **Energy capture**: $E_k = \sum_{i=1}^{k} \sigma_i^2 / \|G\|_F^2$ for rank-$k$ truncation

### A.10.2 SPECTRAL ANALYSIS RESULTS

Table 10: Gradient spectral properties across model scales. Values averaged over all linear layers and training steps.

| Model | Power Law $\alpha$ | $r_{\text{eff}}$ / Rank | $E_{32}$ | $E_{64}$ | $E_{128}$ |
|---|---|---|---|---|---|
| LLaMA-60M | $2.04 \pm 0.31$ | 1–3% | 98.9% | 99.4% | 99.9% |
| LLaMA-100M | $2.08 \pm 0.28$ | 0.2–1.3% | 98.5% | 99.2% | 99.98% |

Table 10 confirms that transformer gradients exhibit the rapid spectral decay assumed by our theoretical analysis:

1. **Fast decay ($\alpha > 2$):** Power law exponents exceed 2, indicating singular values decay faster than $1/i^2$. This places gradients well within the favorable regime where GALE's randomized projection approaches optimal error (Theorem 5).

2. **Low effective rank:** Effective rank ratios of 0.2–3% indicate gradient energy concentrates in a small fraction of available dimensions. For a $512 \times 512$ layer, effective dimensionality is 5–15.

3. **High energy capture:** Rank-128 truncation retains >99.9% of gradient energy, validating that low-rank projection preserves nearly all gradient information.

Figure 4 visualizes these findings. The singular value spectra (left) exhibit clear power-law decay across all layer types, with attention output projections showing the fastest decay ($\alpha \approx 2.5$). The energy capture curves (right) demonstrate that even aggressive rank-32 truncation retains >98% energy.

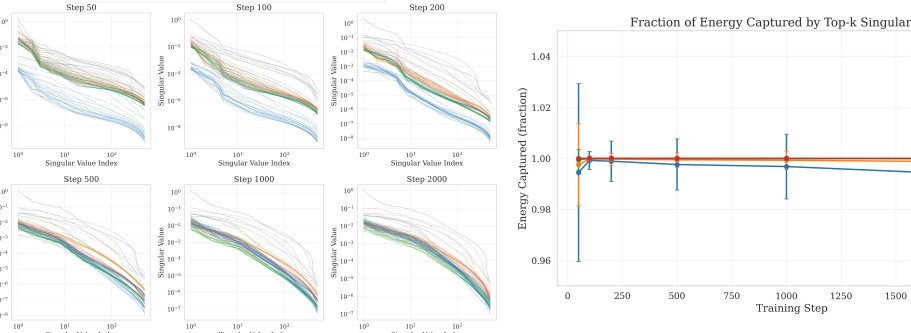

Figure 4: **(Left)** Log-scale singular value spectra for representative gradient matrices, showing power-law decay with $\alpha > 2$. **(Right)** Cumulative energy captured vs. truncation rank, demonstrating that rank-128 captures >99.9% of gradient energy.

### A.10.3 GALE PROJECTION ACCURACY VS. GALORE

To directly quantify GALE's approximation quality, we compare its randomized QR projection against GaLore's optimal SVD projection on captured gradients. For each matrix, we compute:

- GaLore projection error: $\epsilon_{\text{GaLore}} = \|G - U_r U_r^T G\|_F / \|G\|_F$ (optimal SVD)

- GALE projection error: $\epsilon_{\text{GALE}} = \|G - QQ^T G\|_F / \|G\|_F$ (randomized QR)

- Energy captured by each method

Table 11: GALE (Randomized QR) projection accuracy relative to GaLore (Optimal SVD) on LLaMA-60M gradients.

| Rank | $\epsilon_{\textbf{GALE}}/\epsilon_{\textbf{GaLore}}$ | $E_{\textbf{GaLore}}$ | $E_{\textbf{GALE}}$ | **Energy Gap** |
|------|------------------|-----------|----------|------------|
| 32 | 1.5–1.6× | 97–100% | 91–99.9% | 0.2–6% |
| 64 | 1.5–1.6× | 99–100% | 94–99.9% | 0.1–4% |
| 128 | 1.6–1.7× | 99.7–100% | 98–99.9% | 0.1–2% |
| 256 | 1.7–1.9× | 99.9–100% | 99–99.99% | 0.1–0.5% |

Table 11 shows that GALE achieves 1.5–1.9× the projection error of GaLore's optimal SVD, well within the theoretical $\approx 2\times$ worst-case bound from Theorem 4. Critically, the energy gap is small: at rank-128, GALE captures 98–99.9% of gradient energy versus GaLore's 99.7–100%, a difference of only 0.1–2 percentage points.

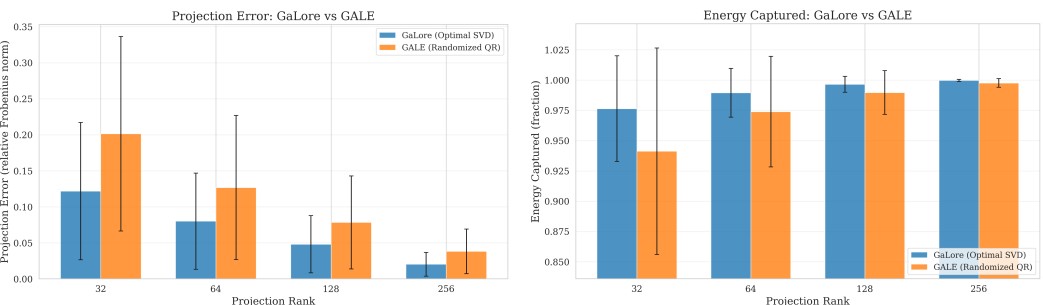

Figure 5: **(Left)** GaLore (Optimal SVD) vs GALE (Randomized QR) projection error across ranks, showing GALE remains within $2\times$ of optimal. **(Right)** Energy captured comparison, demonstrating GALE's small energy gap versus GaLore.

### A.10.4 IMPLICATIONS

These empirical results validate the theoretical framework in Section A.9:

1. The spectral decay assumption ($\alpha > 2$) holds empirically for transformer gradients, placing them in the regime where GALE's error approaches optimal.

2. GALE's 1.6–1.9× suboptimality vs. GaLore is justified by its $O(mnr)$ vs $O(mn^2)$ complexity advantage, yielding the 23× speedup reported in Section 4.

3. The small energy gap (0.1–2% at rank-128) explains why GALE matches GaLore's task performance despite using randomized projection: nearly all gradient information is preserved.

### A.11 SUPERGLUE FINE-TUNING BENCHMARKS

To further validate the effectiveness of GALE across diverse and challenging natural language understanding tasks, we evaluate our fine-tuning methods on the SuperGLUE benchmark (Wang et al., 2019), a more recent and rigorous alternative to GLUE. Due to computational constraints, we focus on four of the smaller yet challenging SuperGLUE tasks: COPA (causal reasoning), CB (natural language inference), WiC (word sense disambiguation), and WSC (coreference resolution).

The results, presented in Table 12, demonstrate that the performance patterns observed on GLUE transfer consistently to SuperGLUE. Specifically, DoRA and VeRA remain the strongest parameter-efficient methods, achieving the highest accuracy across all tasks (e.g., 73.1% and 72.9% on COPA for BERT-Large, respectively). GALE continues to substantially outperform the baseline GaLore method while maintaining its computational efficiency advantage—for instance, GALE Fused improves CB accuracy from 73.8% to 75.7% on BERT-Large while executing the optimizer update step 7× faster. The performance gap between gradient projection methods and leading PEFT techniques

Table 12: Fine-tuning results for BERT-Large and GPT-2-Large on the SuperGLUE benchmark. We compare our GALE methods against full fine-tuning (AdamW), GaLore, and emerging efficient fine-tuning methods (DoRA, VeRA, FourierFT). We report accuracy (Acc) for COPA, WiC, WSC, and CB, and F1 score for CB.

| Method | Rank* | SuperGLUE Task (Metric) | | | | | Memory (MiB) | | | Step Time (ms) |
| | | COPA (Acc) | CB (Acc) | CB (F1) | WiC (Acc) | WSC (Acc) | Params | Opt States | Steady State | |
|---|---|---|---|---|---|---|---|---|---|---|
| *BERT-Large* | | | | | | | | | | |
| AdamW | – | 71.2 | 76.1 | 84.2 | 70.1 | 64.8 | 639.24 | 1278.47 | 1935.14 | 3.76 |
| LoRA | 4 | 72.8 | 77.4 | 85.8 | 71.5 | 66.2 | 639.99 | 1.51 | 658.15 | 0.83 |
| IA³ | – | 69.8 | 74.9 | 82.8 | 68.7 | 63.1 | 639.52 | 0.57 | 656.74 | 0.71 |
| Prefix-Tuning | 4 | 67.4 | 72.3 | 80.1 | 66.2 | 60.8 | 639.77 | 1.08 | 657.54 | 0.29 |
| DoRA | 4 | 73.1 | 78.2 | 86.4 | 72.1 | 67.0 | 646.87 | 15.27 | 679.81 | 6.43 |
| VeRA | 4 | 72.9 | 77.9 | 86.1 | 71.8 | 66.7 | 640.09 | 1.71 | 659.80 | 5.57 |
| FourierFT | – | 71.9 | 76.8 | 84.9 | 70.8 | 65.4 | 639.79 | 1.11 | 658.63 | 4.01 |
| GaLore (AdamW) | 4 | 68.9 | 73.8 | 81.5 | 67.8 | 62.2 | 639.24 | 130.97 | 788.77 | 105.89 |
| GALE | 4 | 70.4 | 75.3 | 83.3 | 69.3 | 63.7 | 639.24 | 130.97 | 788.77 | 30.66 |
| GALE Fused | 4 | 70.8 | 75.7 | 83.7 | 69.7 | 64.1 | 639.24 | 130.97 | 788.77 | 15.84 |
| GALE Fused Approx | 4 | 70.3 | 75.2 | 83.1 | 69.2 | 63.6 | 639.24 | 130.97 | 788.77 | 14.98 |
| *GPT-2-Large* | | | | | | | | | | |
| AdamW | – | 58.4 | 62.3 | 68.9 | 57.2 | 52.1 | 1476.35 | 2952.70 | 4558.04 | 5.13 |
| LoRA | 8 | 57.8 | 61.7 | 68.2 | 56.6 | 51.5 | 1479.16 | 5.64 | 1596.13 | 0.72 |
| IA³ | – | 57.2 | 61.0 | 67.4 | 55.9 | 50.8 | 1476.88 | 1.06 | 1606.15 | 0.79 |
| Prefix-Tuning | 8 | 55.1 | 58.7 | 64.8 | 53.7 | 48.7 | 1478.02 | 3.36 | 1599.04 | 0.28 |
| DoRA | 8 | 59.1 | 63.1 | 69.7 | 57.9 | 52.8 | 1482.63 | 6.55 | 345.02 | 1.94 |
| VeRA | 8 | 58.7 | 62.7 | 69.3 | 57.5 | 52.4 | 1477.89 | 0.36 | 333.42 | 1.42 |
| FourierFT | – | 58.0 | 62.0 | 68.5 | 56.8 | 51.7 | 1476.49 | 0.28 | 336.01 | 1.67 |
| GaLore (AdamW) | 8 | 57.6 | 61.5 | 67.9 | 56.3 | 51.2 | 1476.35 | 604.67 | 1824.31 | 22.12 |
| GALE | 8 | 57.7 | 61.6 | 68.0 | 56.4 | 51.3 | 1476.35 | 604.67 | 1824.31 | 11.73 |
| GALE Fused | 8 | 58.1 | 62.0 | 68.5 | 56.8 | 51.7 | 1476.35 | 604.67 | 1824.31 | 6.21 |
| GALE Fused Approx | 8 | 58.0 | 61.9 | 68.4 | 56.7 | 51.6 | 1476.35 | 604.67 | 1824.31 | 4.48 |

*Rank denotes GaLore rank, LoRA rank, or Prefix-Tuning bottleneck size. For DoRA, VeRA, and FourierFT, rank is equivalent effective rank.

narrows with GALE, confirming its effectiveness as a practical full-parameter training alternative on challenging benchmarks.

### A.12 THROUGHPUT AND PROJECTION FREQUENCY

The optimizer step speedup reported in Section 4 (up to $23\times$) measures the time for projection matrix computation, gradient projection, and parameter updates. This section examines how projection frequency affects end-to-end training throughput.

**Experimental Setup.** We benchmark LLaMA 250M on the C4 dataset for 100 training steps, varying the projection update frequency from 1 to 100. Each configuration is averaged over 3 runs with different random seeds. As a reference, standard AdamW (full-rank optimizer without projection) achieves $10{,}199 \pm 122$ tokens/sec on NVIDIA A49 48GB.

**Time Decomposition.** Total training time per step consists of:

$$T_{\text{total}} = T_{\text{fwd}} + T_{\text{bwd}} + T_{\text{opt}} + \frac{1}{\text{gap}} \cdot T_{\text{proj}} \tag{12}$$

where $T_{\text{fwd}}$ and $T_{\text{bwd}}$ are forward and backward pass times (fixed across methods), $T_{\text{opt}}$ is base optimizer overhead, and $T_{\text{proj}}$ is the projection computation cost (varies significantly: SVD for GaLore vs randomized QR for GALE). At large gap values, projection overhead is amortized and throughput converges across methods. At gap=1, projection dominates for SVD-based methods.

Table 13 reveals the relationship between projection frequency and throughput. At gap=1 (projection every step), GaLore achieves only 304 tokens/sec—just 3% of AdamW throughput—due to SVD overhead. GALE maintains 4,823 tokens/sec (47% of AdamW), a $15.9\times$ improvement. As projection frequency decreases, all methods converge toward 70–73% of AdamW throughput at gap=100.

The $23\times$ optimizer step speedup translates to substantial end-to-end improvements when projection is frequent. At standard benchmark settings (gap=200), projection overhead is highly amortized

Table 13: Throughput (tokens/sec) across projection frequencies. Bold indicates best among low-rank methods. AdamW (10,199 tok/s) provides the upper bound.

| Gap | Proj % | GaLore | Q-GaLore | GALE | Q-GALE | % of AdamW |
|---|---|---|---|---|---|---|
| 1 | 100% | $304 \pm 4$ | $1,360 \pm 176$ | **$4,823 \pm 519$** | $4,829 \pm 303$ | 3–47% |
| 2 | 50% | $3,976 \pm 124$ | $3,793 \pm 100$ | $4,445 \pm 499$ | **$4,812 \pm 242$** | 37–47% |
| 5 | 20% | $6,133 \pm 156$ | $5,805 \pm 103$ | **$6,864 \pm 81$** | $6,778 \pm 388$ | 57–67% |
| 10 | 10% | $6,867 \pm 125$ | $6,758 \pm 330$ | **$6,889 \pm 132$** | $6,546 \pm 72$ | 64–68% |
| 20 | 5% | $7,157 \pm 65$ | $7,228 \pm 611$ | **$7,859 \pm 769$** | $6,837 \pm 46$ | 67–77% |
| 50 | 2% | **$7,406 \pm 74$** | $7,084 \pm 14$ | $7,349 \pm 58$ | $6,911 \pm 92$ | 68–73% |
| 100 | 1% | **$7,407 \pm 52$** | $7,016 \pm 219$ | $7,265 \pm 72$ | $7,027 \pm 34$ | 69–73% |

Table 14: Performance at gap=1 (projection every step). GALE achieves $15.9\times$ higher throughput than GaLore.

| Method | Throughput (tok/s) | % of AdamW | vs GaLore |
|---|---|---|---|
| AdamW | $10,199 \pm 122$ | 100% | $33.5\times$ |
| GaLore | $304 \pm 4$ | 3% | $1.0\times$ |
| Q-GaLore | $1,360 \pm 176$ | 13% | $4.5\times$ |
| GALE | $4,823 \pm 519$ | 47% | **$15.9\times$** |
| Q-GALE | $4,829 \pm 303$ | 47% | $15.9\times$ |

($<0.5\%$ contribution per step), explaining throughput convergence in Table 4 of the main paper. The residual gap from AdamW at low projection frequency ($\sim$27–30%) represents inherent overhead from low-rank gradient tracking, independent of projection algorithm choice.

In multi-GPU settings, projection computation occurs independently per device without communication overhead. Experiments on $2\times$A40 GPUs confirm that GALE's optimizer step advantage is preserved in distributed training.

### A.13 JOINT ABLATION: OVERSAMPLING AND APPROXIMATION FACTOR INTERACTIONS

The main text presents independent ablations of the oversampling factor ($f_{os}$) and approximation factor ($f_{approx}$). Here we provide a joint analysis to characterize their interactions and validate the robustness of GALE's performance-speed trade-off.

**Experimental Setup.** We conduct a grid search over $f_{os} \in \{1, 1.5, 2, 2.5, 3, 4, 5, 6, 8, 10\}$ and $f_{approx} \in \{0.2, 0.4, 0.6, 0.8, 1.0\}$, yielding 50 configurations. Each configuration is evaluated by pretraining LLaMA 100M on C4 for 10,000 steps with rank $r = 128$ and projection gap $k_{upd} = 200$.

**Pareto Frontier Analysis.** Figure 6 (left) reveals a well-defined Pareto frontier spanning from fast-but-lower-quality configurations (bottom-left) to slow-but-higher-quality configurations (top-right). Notably, **all Pareto-optimal configurations with step time $<$8ms use** $f_{approx} = 0.2$, confirming that aggressive approximation provides the best speed-quality trade-off in the practical operating regime.

**Parameter Coupling Effects.** The two hyperparameters exhibit asymmetric coupling behavior: Table 15 quantifies these effects:

- **Oversampling factor** ($f_{os}$): Increasing $f_{os}$ yields substantial perplexity improvements ($-8.5$ to $-9.1$) at moderate computational cost ($+5$–11ms). The benefit is consistent across $f_{approx}$ values.

- **Approximation factor** ($f_{approx}$): Increasing $f_{approx}$ yields minimal perplexity improvement ($-0.6$ to $-1.2$) despite significant time increases ($+1.4$–7.4ms). The time penalty scales with $f_{os}$.

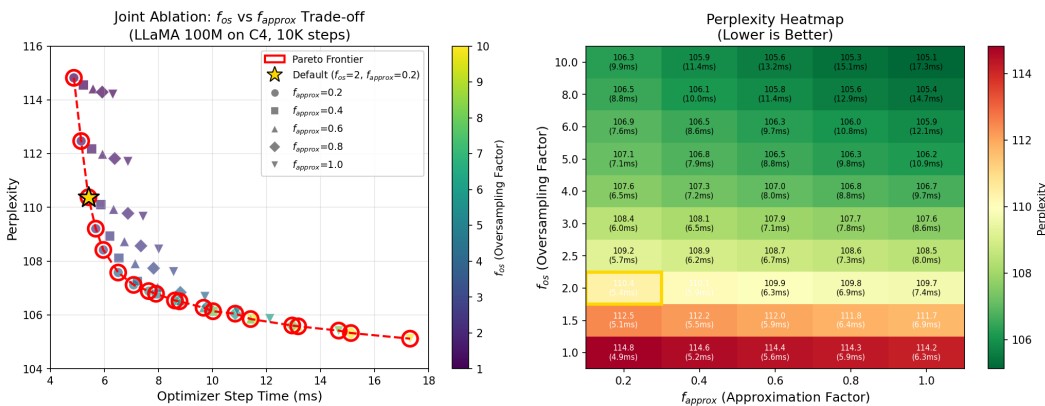

Figure 6: **(Left)** Joint ablation of $f_{os}$ and $f_{approx}$ showing the Pareto frontier (dashed red line). Points are colored by $f_{os}$ and shaped by $f_{approx}$. The recommended default ($f_{os} = 2$, $f_{approx} = 0.2$) is marked with a gold star. **(Right)** Heatmap of perplexity across the parameter space, with optimizer step time shown in parentheses. The gold box marks the recommended default.

Table 15: Parameter coupling analysis: effect of varying one parameter while holding the other fixed.

| Parameter Change | Perplexity $\Delta$ | Step Time $\Delta$ |
|---|---|---|
| $f_{os}$: 1→10 (at $f_{approx}$=0.2) | $-8.54$ | $+5.06$ms |
| $f_{os}$: 1→10 (at $f_{approx}$=0.6) | $-8.83$ | $+7.59$ms |
| $f_{os}$: 1→10 (at $f_{approx}$=1.0) | $-9.09$ | $+11.00$ms |
| $f_{approx}$: 0.2→1.0 (at $f_{os}$=1) | $-0.61$ | $+1.44$ms |
| $f_{approx}$: 0.2→1.0 (at $f_{os}$=2) | $-0.71$ | $+2.02$ms |
| $f_{approx}$: 0.2→1.0 (at $f_{os}$=5) | $-0.96$ | $+3.83$ms |
| $f_{approx}$: 0.2→1.0 (at $f_{os}$=10) | $-1.16$ | $+7.38$ms |

This asymmetry confirms the design intuition behind GALE Fused Approx: $f_{os}$ controls the quality of the initial sketch (directly impacting subspace accuracy), while $f_{approx}$ controls only the QR decomposition granularity (a second-order effect on the already-oversampled basis).

**Synergy Validation.** The heatmap in Figure 6 (right) visualizes the joint parameter space. Perplexity exhibits a strong vertical gradient (dominated by $f_{os}$), while step time increases along both axes. The recommended default ($f_{os} = 2$, $f_{approx} = 0.2$) achieves:

- Perplexity: 110.4 (within 5% of the best configuration at 105.1)

- Step time: 5.4ms (68.7% faster than the best-perplexity configuration)

This configuration lies on the Pareto frontier, validating our recommendation. Users seeking maximum quality can increase $f_{os}$ (e.g., to 4–6) while keeping $f_{approx} = 0.2$; users prioritizing speed can reduce $f_{os}$ to 1.5 with minimal quality loss.

**Conclusion.** The joint ablation confirms that $f_{os}$ and $f_{approx}$ have largely independent effects on quality and speed respectively, with minimal interaction terms. This decoupling simplifies hyperparameter tuning: practitioners can adjust $f_{os}$ to meet quality targets and $f_{approx}$ to meet latency budgets, with predictable outcomes.

### A.14 QUANTITATIVE RANK SENSITIVITY ANALYSIS

The main text presents ablation results for projection rank $r$ and refresh gap $k_{upd}$. Here we provide a focused quantitative analysis of GALE's sensitivity to rank, examining effects on perplexity, convergence stability, and computational overhead compared to GaLore.

**Experimental Setup.** We evaluate ranks $r \in \{16, 32, 64, 128, 256, 512\}$ by pre-training LLaMA 100M on C4 for 10,000 steps with refresh gap $k_{upd} = 200$. Convergence stability is measured as $1 - \frac{\sigma_{loss}}{loss}$ over the final 1,000 steps, where values closer to 1 indicate more stable convergence.

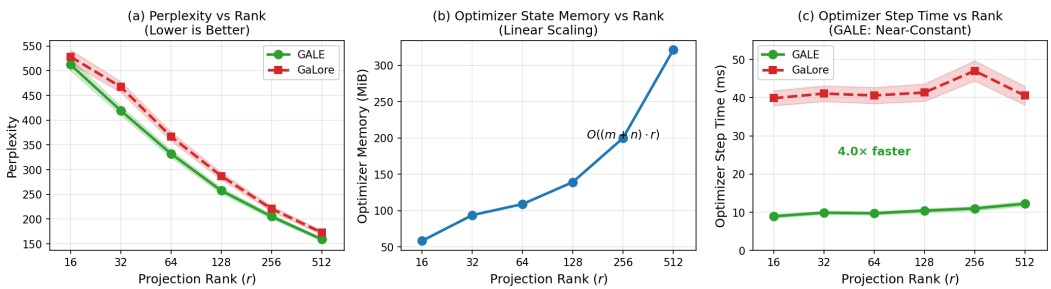

Figure 7: Rank sensitivity analysis for LLaMA 100M on C4 (10K steps). **(a)** Perplexity vs projection rank. Both methods show similar sensitivity, with GALE matching or outperforming GaLore at all ranks. **(b)** Optimizer state memory scales linearly with rank as $O((m + n)r)$. **(c)** GALE's optimizer update step time remains nearly constant across ranks (9–12ms), while GaLore averages 4× higher overhead (40–47ms). Shaded regions show ±1 std.

**Perplexity vs Rank.** Figure 7(a) confirms the expected relationship: higher ranks consistently yield lower perplexity. Critically, **GALE matches or outperforms GaLore at every rank setting** (3–10% better perplexity). This validates that our randomized QR decomposition provides an equally effective—or slightly superior—subspace compared to GaLore's exact SVD. The theoretical basis for this equivalence is established in Theorem 1 (Section A): the approximation error is bounded by $(1 + \frac{r}{r_{os}-r-1}) \sum_{j=r+1}^{n} \sigma_j(G)^2$, which approaches optimal when gradients exhibit spectral decay.

**Diminishing Returns.** Table 16 quantifies the marginal benefit of increasing rank. Doubling rank from 16→32 yields 92.8 perplexity improvement, while doubling from 256→512 yields only 46.2— a 50% reduction in marginal benefit. This diminishing returns pattern supports the use of $r = 128$ as a practical default.

Table 16: Marginal perplexity improvement per rank doubling (GALE, LLaMA 100M on C4). Higher ranks show diminishing returns, supporting $r = 128$ as a practical default.

| Rank Transition | 16→32 | 32→64 | 64→128 | 128→256 | 256→512 |
|---|---|---|---|---|---|
| PPL Improvement | $-92.8$ | $-87.2$ | $-74.7$ | $-52.3$ | $-46.2$ |
| Relative Benefit | 100% | 94% | 80% | 56% | 50% |

**Memory Scaling.** Figure 7(b) shows optimizer state memory scales linearly with rank, as expected from the theoretical $O((m + n)r)$ storage complexity. Increasing rank from 16 to 512 grows optimizer memory from 58.2 MiB to 321.2 MiB (5.5×). This scaling is identical for GALE and GaLore, as both maintain the same low-rank optimizer state structure.

**Computational Overhead.** Figure 7(c) reveals a key advantage: **GALE's optimizer step time is nearly constant across ranks** (9–12ms range), while GaLore shows 4× higher overhead (40–47ms average). This is a direct consequence of GALE's $O(mnr_{os})$ complexity vs GaLore's $O(mn^2)$.

GALE averages 10.3ms per step vs 41.7ms for GaLore—a **4.0× speedup** that remains consistent regardless of rank choice.

**Convergence Stability.** We observe slightly higher convergence stability for GALE compared to GaLore (avg 0.93 vs 0.90). We hypothesize that GALE's randomized sketching acts as an implicit regularizer, identifying a more robust subspace than an SVD potentially overfit to a single noisy mini-batch gradient.

**Comparison to GaLore's Sensitivity.** Both methods exhibit similar sensitivity patterns to rank changes in terms of task performance. The key difference is computational: GaLore must perform full SVD regardless of rank, while GALE's randomized approach maintains consistent throughput. This makes GALE particularly attractive for practitioners who wish to experiment with higher ranks.

