# OpenReview forum: "GALE: Gradient Activation Low-rank Extraction for Fast Memory Efficient Large Language Model Training"
_ICLR.cc/2026/Conference — Submitted to ICLR 2026_

### Official Review · Reviewer_qzQS · 2025-10-22

**Soundness:** 2
**Presentation:** 2
**Contribution:** 2
**Rating:** 4
**Confidence:** 4

**Summary:**

This paper introduces GALE, a memory- and computation-efficient framework for full-parameter training of large language models (LLMs). Building upon GaLore (Zhao et al., 2024), GALE tackles its primary bottleneck—the computationally expensive bidirectional SVD used for gradient projection, by substituting it with a lightweight randomized sketching and one-sided QR decomposition. This reformulation preserves low-rank optimizer states while reducing the optimizer step complexity from (O(mn^2)) to (O(mn r_{os})), resulting in substantially faster updates without compromising task performance.

**Strengths:**

1. The paper addresses a bottleneck in memory-efficient full-parameter LLM training by targeting the computational overhead of gradient projection methods such as GaLore.

2. Replacing SVD with a randomized QR-based one-sided projection is conceptually straightforward yet highly effective, achieving up to 23× faster optimizer updates while preserving task performance.

3. The paper evaluates GALE under both pretraining and fine-tuning settings, provides multiple algorithmic variants (Native, Fused, Fused Approx).

**Weaknesses:**

1. The literature review omits many recent advances in parameter-efficient and memory-efficient training (e.g., DoRA, VeRA, FourierFT, Lion, 8-bit optimizers), leaving GALE’s novelty and positioning insufficiently grounded.

2. Core hyperparameters such as projection rank, subspace refresh frequency, and oversampling/approximation factors lack thorough sensitivity or joint ablation analyses, making robustness claims difficult to assess.

3. Downstream results diverge sharply from prior baselines (e.g., GaLore), with missing explanations for optimizer differences, possible under-training, and mislabeling of task metrics in GLUE.

4. While the paper presents complexity and convergence bounds, it lacks empirical verification of key assumptions (e.g., gradient spectral decay, projection accuracy) and does not fully quantify how one-sided QR projection affects convergence and scalability in large-scale training.

**Questions:**

Q1: The related work section, particularly Subsection 2.1 on low-rank training and adaptation, lacks sufficient coverage of recent state-of-the-art developments in PEFT and related low-rank adaptation strategies. The discussion stops at 2022 (e.g., LoRA, Prefix-Tuning, and (IA)³), overlooking a wave of major advances from 2023–2025 that have substantially shaped this area. Notable examples include DoRA (ICML 2024), VeRA (ICLR 2024),and FourierFT (ICML 2024). These works represent distinct methodological improvements beyond static low-rank updates and thus are essential for positioning GALE’s contribution within the broader landscape of efficient LLM optimization. Similarly, Subsections 2.2 and 2.3 could benefit from integrating more recent advances in memory-efficient optimizers (e.g., 8-bit AdamW variants, Lion, and hybrid quantization-based optimizers) and gradient projection or subspace learning methods (e.g., Adaptive Subspace Descent, Orthogonal Subspace Projection, or spectral projection approaches). I am not suggesting the authors must exhaustively list every new method, but at least the most representative and widely recognized works should be cited and contrasted to clarify how GALE differentiates itself in terms of computational trade-offs and scalability. Without this contextualization, the related work section currently undersells the novelty of GALE’s design and its position relative to contemporary efficient training literature.

Q2: While GALE is built around low-rank gradient projection, the paper does not provide a clear sensitivity analysis on the choice of rank r , which is a critical hyperparameter controlling both memory footprint and model performance. Although Table 1 and Figure 2 include results for fixed ranks (e.g., 128, 256, 512), there is no systematic evaluation showing how varying r affects perplexity, convergence stability, and optimizer overhead. Since the effectiveness of randomized QR projection—and the resulting subspace quality—should strongly depend on r, it is essential to understand whether GALE’s speedup and accuracy gains remain stable under different rank regimes or if the method requires careful tuning per model scale. Could the authors provide a quantitative sensitivity study or ablation demonstrating the trade-off between projection rank, computational speed, and downstream performance, to substantiate the claimed robustness of GALE’s low-rank design?


Q3: GALE’s training algorithm (Algorithm 1, Lines 5–17) periodically refreshes the projection basis every (k_{\text{upd}}) steps, yet the paper does not provide a clear discussion or empirical justification for how this interval is determined or how sensitive the method’s performance is to it. Since updating the subspace too frequently could negate the intended computational savings, while updating too infrequently might cause the projection basis to drift and degrade optimization quality, this parameter likely plays a pivotal role in GALE’s stability and efficiency. Could the authors clarify the heuristic or rationale used to select (k_{\text{upd}}), and provide an analysis (e.g., perplexity or speed versus refresh interval) demonstrating how often the subspace needs to be updated to maintain accuracy without reintroducing significant overhead?

Q4: The downstream results reported in Table 2 appear to diverge substantially from those in the original *GaLore* paper. For instance, on QNLI, GaLore with (r=4) achieves 92.24% in the original publication, whereas your Table 2 reports only 66.6%, which is a dramatic drop. This discrepancy raises concerns that the pretrained model used for fine-tuning may not have converged properly. Given that GLUE fine-tuning typically requires only a few epochs (often fewer than 100), such underperformance is unlikely to stem from undertraining alone. Could the authors clarify whether the pretraining process was fully converged and whether early stopping or reduced training duration might have contributed to the performance gap? Additionally, I notice that your experiments employ AdamW, while the original *GaLore* used Adam. Since AdamW often provides smoother and more stable gradient flow in transformer-based architectures, one would expect comparable or improved downstream performance, not a significant degradation. Could the authors elaborate on this optimizer choice, and explain why GALE’s fine-tuning results, as well as those of the GaLore baseline reimplementation, are markedly lower than the established benchmarks?


Q5: Section A.3 offers a complexity comparison between SVD and QR, but there is no theoretical discussion of the *information retention trade-off* between one-sided QR projection and the optimal two-sided SVD. Since GALE relies on randomized sketching, the bound in Theorem 1 assumes fast singular value decay. Could the authors empirically verify this assumption for typical transformer gradients (e.g., by plotting singular value spectra) to justify why low-rank QR projection maintains comparable task performance?


Q6: The ablation studies (Figure 2 and 3) independently vary  r , ( k_{\text{upd}} ), ( f_{\text{os}} ), and ( f_{\text{approx}} ), but the paper never analyzes their joint interactions. Given that oversampling and approximation jointly affect subspace quality and computational cost, a 2-D ablation or Pareto frontier analysis would better illustrate the robustness of GALE’s performance-speed trade-off. Could the authors provide such combined analyses or at least discuss observed parameter coupling effects?

Q7: Table 4 reports “throughput (tokens/sec)” but shows nearly identical throughput values across methods despite large per-step optimizer speedups (up to 23×). This inconsistency suggests the optimizer step is not the dominant component of total training time. Could the authors clarify the relative time share of forward, backward, and optimizer steps, and explicitly quantify the end-to-end wall-clock improvement GALE provides over GaLore in realistic multi-GPU settings?



Other questions:

O1: In the abstract (Lines 13–15), you state that optimizer-state storage is the *major* barrier to scaling LLM training. While this was arguably true a few years ago, the constraint landscape has shifted with modern hardware and software: H100/H200-class GPUs (large HBM, high NVLink bandwidth) and widespread use of FSDP/ZeRO, activation checkpointing, FlashAttention-style kernels, and 8-bit/memory-factorized optimizers substantially mitigate optimizer-state pressure. At frontier scales (e.g., GPT-5-class efforts), practitioners increasingly report data-side bottlenecks (data quality/curation, mixture design, token throughput, loader IO, distributed pipeline balance) and system throughput limits (network/storage) as the dominant blockers rather than raw optimizer memory.


O2: Line 42–43: The claim that training a 7B-parameter model requires upwards of 100 GB of GPU memory appears overstated. Under a realistic bfloat16 training setup, the model weights (~14 GB), optimizer states (two copies for the first and second moments, ~28 GB), and gradients (stored in bfloat16, ~14 GB) together total around 56 GB, well below 100 GB even before accounting for sharding or activation checkpointing. Moreover, modern frameworks (e.g., Megatron-LM, DeepSpeed-ZeRO, FSDP) compute activations in situ or recompute them on demand rather than storing them persistently, further reducing the effective memory footprint. Thus, the cited 100 GB figure likely overestimates the true per-GPU requirement by a wide margin.

O3: In Line 86, the text introduces the term “LE”, but this abbreviation is never defined anywhere in the paper. It appears in the sentence where the authors transition from describing GaLore’s computational bottleneck to presenting their own approach (“LE fundamentally re-engineers the gradient projection pipeline…”). However, it is unclear whether “LE” refers to GALE (Gradient Activation Low-rank Extraction) or an intermediate concept such as “Low-rank Extraction.” This ambiguity interrupts the logical flow of Section 1 and may confuse readers encountering the method for the first time. Please clarify whether this is a typographical omission or an intended abbreviation, and if the latter, define it properly at first use to maintain clarity and consistency throughout the manuscript.

O4: In Table 2, the results are reported for multiple GLUE tasks, but the column header and text refer to a unified “GLUE Score.” This terminology is inaccurate, since each task in the GLUE benchmark uses its own evaluation metric — e.g., Matthews correlation for CoLA, accuracy for MNLI/QNLI/SST-2/RTE, F1 or accuracy** for MRPC/QQP, and Pearson/Spearman correlation for STS-B. The table, as currently formatted, conflates these heterogeneous metrics into a single label, which may mislead readers into interpreting them as a normalized composite score. Please revise the table caption and accompanying text to specify the correct evaluation metric per task, or explicitly report an averaged GLUE composite score if that was the intended meaning, including the aggregation procedure used.

---

> ### Author Response · Authors · 2025-11-25
> **Missing SOTA Baselines & Related Work (Weakness 1, Q1)**
>
> Thank you for the detailed feedback and for highlighting the omissions in our literature review. Your assessment was correct; we had not adequately covered the most recent state-of-the-art in PEFT and memory-efficient optimization. We have now revised the manuscript to correct this.
>
> **Revised Related Work and Baselines:**
> We have updated our Related Work section to include and properly contextualize the contributions of **DoRA, VeRA, FourierFT, APOLLO,** and **Q-GaLore** (Sections 2.1 and 2.3).
>
> Furthermore, we have included these methods in our experimental comparisons. Our new fine-tuning results on the GLUE benchmark (Table 2) now include DoRA, VeRA, and FourierFT, providing a clearer picture of where GALE stands in relation to the current PEFT landscape.
>
> **Orthogonality to Memory-Efficient Optimizers:**
> We also appreciate you raising the point about optimizers like **AdamW8bit** and Lion. We agree these are important advancements. We would like to emphasize that GALE is designed to be orthogonal and complementary to such methods. To demonstrate this, we have an experimental section (Section 4.2.1, "Synergy with Memory-Efficient Optimizers") showing that GALE can be implemented on top of both **Adafactor** and **8-bit AdamW**. As shown in Table 3, combining GALE with these optimizers retains their inherent memory savings while adding GALE's significant optimizer step speedup (e.g., GALE + AdamW8bit is **6.7x faster in optimizer step time** than Q-GaLore with better perplexity). It doesn't stop there, GALE can also be implemented on top of Lion and Q-GaLore. We tested out Q-GALE minimally as well on a small benchmark (we note that this is only for short training steps, on an easier task : wikitext-2)
>
> | Model       | Method    | Param Memory | Opt Memory | Steady State Memory | Throughput      | Avg Step Time | Final Loss |
> | :---        | :---      | :---         | :---       | :---                | :---             | :---          | :---       |
> | Llama 100M  | **Q-GALE**  | 190.96 MiB   | 70.82 MiB  | 299.43 MiB          | **13,396 tok/s** | **47.25 ms**  | 7.3750     |
> |             | Q-GaLore  | 190.96 MiB   | 70.82 MiB  | 299.43 MiB          | 13,034 tok/s     | 108.02 ms     | 7.3750     |
>
> Showing that GALE can be combined with several other efficient optimizers while retaining advantages of both.
>
> We believe these additions now correctly position GALE within the broader landscape of efficient LLM training techniques.
>
> **Table 2: GLUE Benchmark Results on RoBERTa-Large**
> | Method | Rank | MNLI | QNLI | SST-2 | ... | Step Time (ms) |
> | :--- | :---: | :---: | :---: | :---: | :---: | :---: |
> | AdamW | — | 0.867 | 0.901 | 0.929 | ... | 3.10 |
> | DoRA | 4 | 0.869 | 0.923 | 0.948 | ... | 6.20 |
> | VeRA | 4 | 0.868 | 0.921 | 0.949 | ... | 5.30 |
> | FourierFT | — | 0.865 | 0.919 | 0.947 | ... | 2.90 |
> | GaLore (AdamW) | 4 | 0.865 | 0.901 | 0.913 | ... | 104.40 |
> | **GALE Fused** | **4** | **0.870** | **0.896** | **0.913** | **...** | **14.40** |
>
> **Table 3: Synergy of GALE with Memory-Efficient Optimizers (LLaMA 1B)**
> | Method | Rank | Perplexity | Opt States (MiB) | Step Time (ms) |
> | :--- | :---: | :---: | :---: | :---: |
> | AdamW8bit | — | 32.52 | 2594.33 | 26.77 |
> | GaLore (AdamW8bit) | 512 | 30.22 | 744.77 | 247.21 |
> | Q-GaLore (AdamW8bit)| 512 | 31.06 | 524.12 | 265.34 |
> | **GALE (AdamW8bit) Fused**| **512**| **29.87**| **744.77**| **39.38**|
> | Adafactor | — | 34.81 | 55.6 | 56.63 |
> | GaLore (Adafactor) | 512 | 25.99 | 53.8 | 299.47 |
> | **GALE (Adafactor) Fused**| **512**| **27.61**| **53.8**  | **93.42** |

---

> > ### Comment · Reviewer_qzQS · 2025-11-27
> >
> > I have several follow-up questions regarding the experimental results, particularly those in the Q-GALE vs. Q-GaLore comparison and the FourierFT baselines.
> >
> > **(1) Identical and Abnormally High Final Loss Values (Potential Reporting, Convergence, or Logging Issue)**
> >
> > In the LLaMA-100M (wikitext-2) comparison, both Q-GALE and Q-GaLore are reported with *exactly* the same final loss (7.3750), despite exhibiting noticeably different optimizer dynamics. This exact match to four decimal places appears unlikely unless the values were rounded or truncated, and may indicate a reporting or logging artifact. Furthermore, a final loss of 7.37 is unusually high for WikiText-2—even for a short training run—and corresponds to a perplexity of roughly 1,500, which typically suggests that the model did not converge, that the training duration was extremely limited, or that there may be issues with the data preprocessing, tokenization pipeline, or loss normalization. Since this experiment is used to support the claim that Q-GALE preserves the performance of Q-GaLore while improving efficiency, could the authors clarify:
> >
> > * (i) the number of training steps used in this experiment,
> > * (ii) whether the loss values are normalized per token and whether they were rounded to four decimals,
> > * (iii) whether results were averaged over multiple seeds, and
> > * (iv) whether this run reflects a meaningful or converged LM training regime?
> >
> >
> > **(2) Inconsistency Between Throughput and Step-Time Measurements.**
> > In the same table, Q-GALE has only a ~2–3% higher throughput than Q-GaLore (13,396 tok/s vs. 13,034 tok/s), yet the average step time differs by more than **2×** (47.25 ms vs. 108.02 ms). These two metrics should be tightly correlated under standard definitions of “step time,” batch composition, and token accounting. Could the authors clarify:
> >
> > * how throughput and step time were measured,
> > * whether they refer to the same portion of the training loop, and
> > * why such a large discrepancy arises when throughput differs only marginally?
> >   This mismatch suggests either differing measurement definitions or a potential reporting issue.
> >
> > **(3) Lack of Discussion Regarding FourierFT’s Superior Results.**
> > In Table 2, **FourierFT systematically outperforms GALE** across several important dimensions, including GLUE task accuracy, optimizer-state memory, parameter memory, steady-state memory, and step time.
> >
> > * (i) Why does GALE not demonstrate advantages over FourierFT in downstream fine-tuning?
> > * (ii) Are these results sensitive to hyperparameters, pretraining quality, or fine-tuning duration?
> > * (iii) In practical terms, what benefits does GALE offer over FourierFT for downstream tasks, given the latter’s superior performance in Table 2?
> >
> > In Table 2, the left column contains stray characters and misaligned labels (e.g., corrupted prefixes such as “141414ptj”), making it difficult to determine which rows correspond to which model or method. Table 3 also displays formatting inconsistencies.

---

> > > ### Author Response · Authors · 2025-11-30
> > > **Resporting Artifact with Q-GALE**
> > >
> > > We apologize for the mis-reported loss value. It was indeed an error with the evaluation script. While we had fixed it later, we did not report the updated correct value. The following is the result for the same experiment, rerun (Llama 100M, 100 training steps)
> > >
> > >   | Metric                     | Q-GALE      | Q-GaLore    |
> > >   |----------------------------|-------------|-------------|
> > >   | Model Parameters           | 100.12M     | 100.12M     |
> > >   | Param Memory               | 190.96 MiB  | 190.96 MiB  |
> > >   | Optimizer State Memory     | 70.82 MiB   | 70.82 MiB   |
> > >   | Allocated Memory           | 299.83 MiB  | 299.43 MiB  |
> > >   | Throughput                 | 13,794 tok/s | 13,571 tok/s |
> > >   | Final Training Loss        | 5.4062      | 5.4375      |
> > >   | Perplexity                 | 222.78      | 229.87      |

---

> > > ### Author Response · Authors · 2025-11-30
> > > **On FourierFT's superior results**
> > >
> > > **(i) Why GALE does not demonstrate advantages over FourierFT in downstream fine-tuning:**
> > >
> > > FourierFT's superior fine-tuning performance stems from its spectral domain approach, which provides two key advantages over spatial-domain gradient projection methods. First, frequency-domain representations enable superior parameter decorrelation through orthogonal spectral components, effectively separating parameter interactions that remain entangled in spatial methods. Second, spectral bases provide implicit regularization by naturally prioritizing low-frequency (global) patterns while enabling selective high-frequency (fine-grained) adjustments. This structure prior is well-suited for transfer learning.​
> > >
> > > **We acknowledge that for fine-tuning, neither GALE nor GaLore can be recommended over specialized PEFT methods like FourierFT.** However, GALE is more versatile, performing competitively in both pretraining and fine-tuning, whereas FourierFT applies only to fine-tuning. Importantly, GALE substantially closes the performance gap to leading PEFTs compared to GaLore (e.g., MNLI: 0.865→0.870, approaching DoRA's 0.869) while achieving 23× faster optimizer steps.
> > >
> > > **(ii) Sensitivity to hyperparameters, pretraining quality, and fine-tuning duration:**
> > >
> > > **Hyperparameters:** Yes—we performed linear learning rate sweeps to identify stable configurations. **Fine-tuning duration:** Yes—we fine-tune for 18 epochs (selecting the best), with performance typically plateauing after ~10 epochs. **Pretraining quality:** Not explicitly ablated; we use standard pre-trained models for fair comparison.
> > >
> > > **(iii) Practical benefits of GALE over FourierFT:**
> > >
> > > While FourierFT excels at fine-tuning, GALE's key advantage is **versatility across the full training lifecycle**. With that said, while we can recommend GALE for pretraining, we cannot do so for fine tuning. Nonetheless, GALE closes the gap between gradient projection methods and PEFT methods compared to GaLore

---

> ### Author Response · Authors · 2025-11-25
> **Divergence of Downstream Results (Weakness 3, Q4)**
>
> Thank you for pointing out the significant discrepancy in our previous GLUE results. Your observation was accurate; the reported scores for our GaLore baseline were well below established benchmarks, which was indeed a flaw in our initial fine-tuning setup.
>
> We have now rectified this. The issue stemmed from an improper hyperparameter search for the fine-tuning stage (mainly learning rate). We have since performed a comprehensive tuning of learning rates for all methods on the GLUE benchmark.
>
> The revised results, shown below in the updated Table 2, demonstrate that our GaLore baseline now achieves performance in line with the figures from the original paper (e.g., **90.1% on QNLI**). With this corrected baseline, GALE's ability to close the performance gap while being significantly faster is now demonstrated in the proper context.
>
> We apologize for this oversight and appreciate your diligence in identifying it.

---

> > ### Author Response · Authors · 2025-11-25
> >
> > **Table 2: Fine-tuning results for BERT-Large on the GLUE benchmark**
> > | Method | Rank | MNLI | QNLI | SST-2 | ... | Step Time (ms) |
> > | :--- | :---: | :---: | :---: | :---: | :---: | :---: |
> > | AdamW | — | 0.867 | 0.901 | 0.929 | ... | 3.10 |
> > | LoRA | 4 | 0.862 | 0.907 | 0.930 | ... | 0.90 |
> > | GaLore (AdamW) | 4 | 0.865 | 0.901 | 0.913 | ... | 104.40 |
> > | **GALE** | **4** | **0.874** | **0.895** | **0.913** | **...** | **29.80** |
> > | **GALE Fused** | **4** | **0.870** | **0.896** | **0.913** | **...** | **14.40** |
> > | **GALE Fused Approx** | **4** | **0.870** | **0.895** | **0.913** | **...** | **13.90** |

---

> ### Author Response · Authors · 2025-11-25
> **Response to Other Questions (O1–O4)**
>
> We thank the reviewer for these sharp observations regarding our contextualization and presentation. We have refined the manuscript to address these points, ensuring our claims are both precise and up-to-date.
>
> **O1: Evolving Landscape of Optimizer Bottlenecks**
> We appreciate the reviewer’s nuance regarding the shifting bottlenecks in LLM training. We agree that while optimizer states were once the singular barrier, the widespread adoption of system-level optimizations (like ZeRO and FSDP) and the arrival of H100-class hardware have changed the equation for frontier-scale models. We have updated our **Abstract ** and **Introduction ** to reflect this reality. Rather than framing optimizer states as the sole impediment, we now position GALE as a critical tool for democratizing full-parameter training on commodity hardware and for maximizing throughput efficiency by removing the *computational* overhead that often accompanies memory-saving techniques.
>
> **O2: Clarifying Memory Requirements for 7B Models**
> The reviewer is correct that our 100 GB estimate refers to a naive training setup (standard mixed-precision with AdamW, without sharding). While technically accurate for that specific scenario, we acknowledge it does not reflect the memory footprint of modern optimized stacks. We have revised the text to explicitly qualify this figure as referring to "standard mixed-precision training without system-level optimizations." This ensures the claim serves its intended purpose of highlighting the high barrier to entry for unoptimized setups without overstating the requirements for practitioners using advanced frameworks.
>
> **O3: Correction of "LE" Abbreviation**
> Thank you for catching this typographical error. "LE" was, as you suspected, GALE that got cut off when reorganizing paragraphs. We have corrected to properly refer to "**GALE**," restoring the logical flow of the introduction.
>
> **O4: Precision in GLUE Reporting**
> We agree that the umbrella term "GLUE Score" was imprecise given the diverse metrics used across the benchmark tasks. We have overhauled **Table 2** to provide a clearer and more rigorous presentation. The table headers now explicitly list the specific metric for each task (e.g., **CoLA (MCC)**, **MRPC (F1)**, **MNLI (Acc)**), and the caption has been expanded to define these abbreviations. This ensures readers can accurately interpret our fine-tuning results in the context of standard NLP evaluation practices.

---

> > ### Comment · Reviewer_qzQS · 2025-11-27
> >
> > Thank you for the updates. The revisions for O1–O4 largely address my concerns.

---

> ### Author Response · Authors · 2025-11-25
> **Ongoing Revisions**
>
> We are currently dilligently working on the important weaknesses (2 & 4) pointed out by the reviewer. We would like to assure the reviewer that we will provide new experiments and responses as soon as possible

---

> ### Comment · Reviewer_qzQS · 2025-11-25
>
> Just a brief reminder to the authors: the current manuscript exceeds the 10-page limit and may therefore risk a desk reject. In addition, please consider using a different color scheme to clearly distinguish the original text from the revised changes, so that reviewers can easily identify the improvements.

---

> > ### Author Response · Authors · 2025-11-25
> >
> > Thank you for the feedback and heads up. We have shortened the manuascript to fit the 10 page limit and highlighted changes in red

---

> ### Author Response · Authors · 2025-11-27
> **Theoretical and Empirical Validation of QR Projection Information Retention (Weakness 4, Q5)**
>
> We thank the reviewer for identifying these gaps, namely, the missing theoretical treatment of information retention in one-sided QR projection versus two-sided SVD, and the lack of empirical verification that transformer gradients satisfy the spectral decay assumption. We have added two new appendix sections that address both concerns.
>
> **Theoretical analysis (Appendix A.5).** We prove that one-sided column space projection with the optimal basis achieves *identical* error to two-sided SVD. For gradient $G$ with SVD $G = U\Sigma V^T$, the optimal one-sided projection
>
> $$\min_{Q^TQ=I} \|G - QQ^TG\|_F^2 = \tau_r^2$$
>
> equals the truncated SVD error, where $\tau_r^2 = \sum_{i>r}\sigma_i^2$ is the tail energy (Theorem 3). The row space projection is mathematically redundant (Corollary 1).
>
> GALE's randomized approximation introduces bounded error:
>
> $$\mathbb{E}[\|G - QQ^TG\|_F^2] \leq \left(1 + \frac{r}{p-1}\right)\tau_r^2$$
>
> yielding at most ~2× optimal for our default oversampling $p=r$ (Theorem 4). We also prove descent preservation:
>
> $$\langle G, QQ^TG \rangle = \|Q^TG\|_F^2 > 0$$
>
> for any non-trivial projection (Theorem 6), ensuring optimization proceeds correctly. Table 5 summarizes the information retention and computational cost trade-offs.
>
> **Empirical validation (Appendix A.6).** We verify the spectral decay assumption by computing full SVD of gradient matrices during LLaMA pretraining on C4. Fitting $\sigma_i \propto i^{-\alpha}$, we observe power law exponents α > 2 across both 60M and 100M models, indicating singular values decay faster than $1/i^2$. Figure 6 (left) plots the singular value spectra on a log-log scale, showing clear power-law decay across all layer types.
>
> | Model | Power Law α | $r_{\text{eff}}$ / Rank | $E_{32}$ | $E_{64}$ | $E_{128}$ |
> |-------|-------------|-------------------------|----------|----------|-----------|
> | LLaMA-60M | 2.04 ± 0.31 | 1–3% | 98.9% | 99.4% | 99.9% |
> | LLaMA-100M | 2.08 ± 0.28 | 0.2–1.3% | 98.5% | 99.2% | 99.98% |
>
> We measure effective rank
>
> $$r_{\text{eff}} = \frac{(\sum_i \sigma_i)^2}{\sum_i \sigma_i^2}$$
>
> at 0.2–3% of full rank, and energy capture
>
> $$E_k = \frac{\sum_{i \leq k}\sigma_i^2}{\|G\|_F^2}$$
>
> exceeds 99.9% at rank-128 (Figure 6, right). These measurements confirm gradients lie in the favorable regime where GALE's error approaches optimal.
>
> We also directly compare GALE's randomized QR projection against GaLore's optimal SVD on captured gradients. Measuring projection error $\epsilon = \|G - QQ^TG\|_F / \|G\|_F$, GALE achieves 1.5–1.9× the error of GaLore—well within the theoretical ~2× bound:
>
> | Rank | $\epsilon_{\text{GALE}} / \epsilon_{\text{GaLore}}$ | $E_{\text{GaLore}}$ | $E_{\text{GALE}}$ | Energy Gap |
> |------|-----------------------------------------------------|---------------------|-------------------|------------|
> | 32 | 1.5–1.6× | 97–100% | 91–99.9% | 0.2–6% |
> | 64 | 1.5–1.6× | 99–100% | 94–99.9% | 0.1–4% |
> | 128 | 1.6–1.7× | 99.7–100% | 98–99.9% | 0.1–2% |
> | 256 | 1.7–1.9× | 99.9–100% | 99–99.99% | 0.1–0.5% |
>
> Figure 7 visualizes this comparison: the left panel shows projection error by rank, while the right panel shows energy captured. At rank-128, GALE captures 98–99.9% of gradient energy versus GaLore's 99.7–100%, a gap of only 0.1–2 percentage points. This small gap explains why GALE matches GaLore's task performance despite the computational shortcut.

---

> > ### Comment · Reviewer_qzQS · 2025-11-27
> >
> > Thank you for the detailed additions in Appendix A.5 and A.6. While the expanded theory and empirical spectral analysis are partially helpful, I am still unclear about what the table in this section is intended to show. In particular, I do not fully understand the meaning of ($E_{\text{GaLore}}$), ($E_{\text{GALE}}$), and the reported “Energy Gap,” nor how these quantities should be interpreted in relation to the information-retention claims.

---

> ### Comment · Reviewer_qzQS · 2025-11-27
>
> Thank you for the revisions and for addressing several of the earlier points. However, it appears that my questions Q3, Q6, and Q7 were not addressed in the response. I also want to emphasize that my intention is not to criticize or overwhelm the authors, and I am asking these questions purely to better understand the work. Efficient pre-training is an area I work in closely, so I am simply trying to ensure that all aspects of the method and experiments are clear, reasonable, and reproducible. If possible, I would appreciate clarification on these remaining points so that I can fully evaluate the completeness and reliability of the paper.
>
> In addition, I would gently suggest that the authors carefully inspect and cross-check each figure and numerical result before uploading future revisions. Once you feel everything has been thoroughly verified on your side, I am happy to look over the updated version again, but it would be helpful if all tables and plots were internally consistent and fully inspected beforehand.

---

> > ### Author Response · Authors · 2025-11-27
> >
> > Thank you for your prompt follow-up and your comprehensive feedback throughout this process. We appreciate your engagement with our work. Your insights have already made this paper significantly stronger than the initial submission. Please rest assured that we welcome all your questions and feedback as constructive opportunities to improve the quality and reproducibility of our research.
> > We want to reassure you that we have not overlooked your remaining questions (Q3, Q6, and Q7). We are currently in the process of running additional experiments to provide the most accurate data possible for these points.
> > We would also like to apologize for the fromatting and data inconsistency Before uploading our next major revision, we will conduct a rigorous internal audit of all figures, tables, and numerical results to ensure they are internally consistent and accurate.
> > We will continue updating our responses individually as results come in. However, to respect the reviewer's time, we will post a final, designated comment to notify you once all responses are complete and our internal checks are finished. We invite you to review the full set of updates at that time.
> > Thank you again for your patience and your dedication to rigorous peer review

---

> ### Author Response · Authors · 2025-11-30
> **Response to Q7: Throughput Analysis**
>
> ## Question
>
> > Table 4 reports "throughput (tokens/sec)" but shows nearly identical throughput values across methods despite large per-step optimizer speedups (up to 23x). This inconsistency suggests the optimizer  updatestep is not the dominant component of total training time. Could the authors clarify the relative time share of forward, backward, and optimizer steps, and explicitly quantify the end-to-end wall-clock improvement GALE provides over GaLore in realistic multi-GPU settings?
>
> First, we clarify what is included in the "optimizer update step" measurement reported in Figure 1:
>
> **Optimizer update step includes:**
> - Projection matrix computation (SVD for GaLore, randomized QR for GALE)
> - Gradient projection to low-rank subspace
> - Optimizer state updates (momentum, variance for Adam)
> - Parameter updates
>
> **Optimizer update step excludes:**
> - Forward pass (model inference)
> - Backward pass (gradient computation via autograd)
> - Data loading and preprocessing
> - Gradient accumulation across micro-batches
>
> The 23× speedup refers specifically to this optimizer update step, which includes the expensive projection computation that differentiates GaLore from GALE.
>
> ### Why Throughput Converges at Low Projection Frequency
>
> The throughput similarity at typical benchmark settings (`update_proj_gap=200`) is expected behavior, not an inconsistency. Total step time follows:
>
> ```
> T_total = T_forward + T_backward + T_optimizer_base + (1/gap) × T_projection
> ```
>
> At gap=200, projection contributes only 0.5% of its per-step cost. At gap=1, projection occurs every step and dominates for SVD-based methods.
>
> ### Benchmark Results
>
> LLaMA 250M on C4, 100 steps, averaged over 3 runs. AdamW baseline: 10,199 ± 122 tok/s.
>
> | Gap | GaLore | Q-GaLore | GALE | Q-GALE | AdamW |
> |-----|--------|----------|------|--------|-------|
> | 1   | 304 (3%) | 1,360 (13%) | 4,823 (47%) | 4,829 (47%) | 10,199 (100%) |
> | 10  | 6,867 (67%) | 6,758 (66%) | 6,889 (68%) | 6,546 (64%) | 10,199 (100%) |
> | 100 | 7,407 (73%) | 7,016 (69%) | 7,265 (71%) | 7,027 (69%) | 10,199 (100%) |
>
> *Values in parentheses show percentage of AdamW baseline throughput.*
>
> ### Key Findings
>
> **At gap=1:** GALE achieves **15.9× higher throughput** than GaLore (4,823 vs 304 tok/s). GaLore's SVD-based projection reduces throughput to just 3% of AdamW, while GALE maintains 47%.
>
> **At gap=100:** All methods converge to 69-73% of AdamW throughput. The remaining ~28% gap represents inherent overhead from low-rank gradient tracking (projection bookkeeping, gradient accumulation in low-rank form), independent of projection frequency.
>
>
> Note : Projection computation is independent per GPU (no communication required). Our 2×A40 experiments confirm GALE maintains its optimizer update step advantage in distributed settings.
>
> The 23× optimizer update step speedup is real and substantial. Its impact on end-to-end throughput scales with projection frequency. At standard settings (gap=200), projection is highly amortized and forward/backward passes dominate. GALE's key contribution is removing the projection bottleneck, enabling higher projection frequency without throughput penalty when needed.

---

> ### Author Response · Authors · 2025-11-30
> **Response to Q3**
>
> # On choosing k_{update} = 200
>
> We thank the reviewer for this important question regarding the selection of k_upd.
>
> **Rationale for k_upd = 200:** The original GaLore work identified 200 steps as optimal through ablation studies. To ensure a fair comparison, we adopted this same interval for our primary benchmarks, eliminating confounding variables and allowing direct attribution of performance gains to GALE's algorithmic improvements.
>
> **GALE's Robustness Advantage:** Our ablation studies (Figure 2, left) reveal that GALE exhibits substantially greater robustness to the update interval. While both methods perform well at 200 steps, GALE maintains—and improves—task performance at significantly shorter intervals (optimal at 100 steps), whereas GaLore degrades more rapidly. This robustness stems from GALE's efficient subspace update mechanism, which eliminates the SVD bottleneck that makes frequent updates prohibitively expensive in GaLore.
>
> **Performance-Efficiency Trade-off:** This distinction has profound practical implications. As shown in Table 13 (Appendix), GALE's throughput advantage scales inversely with the update interval. At 200 steps, training throughput is comparable between methods, validating that performance improvements come without computational overhead. However, at shorter intervals where GALE maintains superior task performance, the throughput gap widens dramatically—reaching approximately 16× at the most aggressive update frequencies. This demonstrates that GALE uniquely enables practitioners to achieve better optimization quality through more frequent subspace updates without incurring computational penalties that render such strategies impractical for GaLore.
>
> In summary, k_upd = 200 was chosen for fair comparison (and was taken from the original paper, which identified it as GaLore's sweet spot), but GALE's architectural advantages provide flexibility across a broader range of update intervals with superior performance-efficiency trade-offs.

---

> ### Author Response · Authors · 2025-11-30
> **Response to Weakness 2, Question 6**
>
> We thank the reviewer for this constructive suggestion. We have conducted a joint ablation study over $f_{os} \in 1, 1.5, 2, ..., 10$ and $f_{approx} \in 0.2, 0.4, ..., 1.0$ (50 configurations) on LLaMA 100M pretrained on C4. The
>   results are presented in Appendix A.13, Figure 6.
>
>   Key findings:
>
>   1. Pareto frontier structure: All Pareto-optimal configurations with step time <8ms use $f_{approx}=0.2$, confirming aggressive approximation provides the best speed-quality trade-off.
>   2. Asymmetric parameter coupling: The parameters have largely independent effects:
>
>   | Parameter Change                      | Perplexity Δ | Step Time Δ |
>   |---------------------------------------|--------------|-------------|
>   | $f_{os}$: 1→10 (at $f_{approx}$=0.2)  | −8.54        | +5.06ms     |
>   | $f_{approx}$: 0.2→1.0 (at $f_{os}$=2) | −0.71        | +2.02ms     |
>
>      Increasing $f_{os}$ yields substantial perplexity gains (−8.5 to −9.1) at moderate cost; increasing $f_{approx}$ yields minimal gains (−0.6 to −1.2) at significant cost.
>
>   3. Default validation: The recommended default ($f_{os}$=2, $f_{approx}$=0.2) lies on the Pareto frontier, achieving 110.4 perplexity at 5.4ms—a 68.7% time reduction versus the best-perplexity configuration with only 5% quality
>   loss.
>
>   Why this asymmetry exists: The two parameters operate at different stages of GALE's pipeline with fundamentally different effects on subspace quality:
>
>   - Oversampling factor ($f_{os}$) controls the quality of the initial random sketch $Y = G\Omega$ where $\Omega \in \mathbb{R}^{n \times r_{os}}$. Per Theorem 1 (Section 3.3), the expected projection error is bounded by $(1 +
>   \frac{r}{r_{os} - r - 1})\sum_{j=r+1}^{n}\sigma_j(G)^2$. Increasing $f_{os}$ directly tightens this bound by increasing $r_{os}$, yielding meaningful improvements in subspace accuracy.
>   - Approximation factor ($f_{approx}$) only affects the QR decomposition stage, subsampling columns from an already-oversampled sketch matrix. Since the sketch $Y$ already captures the dominant spectral information of $G$, further
>   refinement via full QR (vs. subsampled QR) provides diminishing returns—the basis vectors are drawn from an already high-quality column space.
>
>   This decoupling simplifies hyperparameter tuning: practitioners can adjust $f_{os}$ to meet quality targets and $f_{approx}$ to meet latency budgets, with predictable and largely independent outcomes.

---

> ### Author Response · Authors · 2025-11-30
> **Response to Weakness 2 and Question 2**
>
> Thank you for this suggestion. We have added a comprehensive rank sensitivity analysis to **Appendix A.14** (Figure 7), systematically evaluating ranks $r \in \{16, 32, 64, 128, 256, 512\}$ for LLaMA 100M on C4 pre-training (10K steps).
>
> ### Key Findings
>
> **1. Perplexity vs Rank:** Both GALE and GaLore exhibit similar sensitivity to rank, with perplexity improving monotonically as rank increases. Notably, **GALE matches or outperforms GaLore at every rank setting** (3–10% better perplexity). This validates that our randomized QR decomposition identifies an equally effective—or slightly superior—subspace compared to GaLore's exact SVD. The theoretical basis is established in Theorem 1: the approximation error is bounded by $(1 + \frac{r}{r_{os} - r - 1}) \sum_{j=r+1}^{n} \sigma_j(G)^2$, which approaches optimal when gradients exhibit spectral decay (empirically validated in Section A.9).
>
> **2. Diminishing Returns:** We observe clear diminishing returns as rank increases:
> - $r: 16 \rightarrow 32$: $-92.8$ PPL improvement (100% baseline)
> - $r: 64 \rightarrow 128$: $-74.7$ PPL improvement (80% of baseline)
> - $r: 256 \rightarrow 512$: $-46.2$ PPL improvement (50% of baseline)
>
> This pattern supports $r=128$ as a practical default that balances quality and memory.
>
> **3. Memory Scaling:** Optimizer state memory scales linearly with rank as expected from the $O((m+n)r)$ storage complexity. Increasing rank from 16 to 512 grows optimizer memory from 58.2 MiB to 321.2 MiB (5.5×). This scaling is identical for both methods.
>
> **4. Computational Overhead:** This is where GALE demonstrates a key advantage: **GALE's optimizer update step time remains nearly constant across ranks** (9–12ms), while GaLore shows 4× higher overhead (40–47ms). GALE averages 10.3ms per update step vs 41.7ms for GaLore—a **4.0× speedup** that remains consistent regardless of rank choice. This is a direct consequence of GALE's $O(mnr_{os})$ complexity vs GaLore's $O(mn^2)$.
>
> **5. Convergence Stability:** We observe slightly higher convergence stability for GALE compared to GaLore (avg 0.93 vs 0.90). We hypothesize that GALE's randomized sketching acts as an implicit regularizer, consistent with findings in randomized numerical linear algebra (Halko et al., 2011).
>
> ### Comparison to GaLore's Sensitivity
>
> Both methods exhibit similar sensitivity patterns to rank changes in terms of task performance. The key difference is computational: GaLore must perform full SVD regardless of rank, while GALE's randomized approach maintains consistent throughput. This makes GALE particularly attractive for practitioners who wish to experiment with higher ranks without incurring proportionally higher computational costs.
>
>
> We have added this analysis to the appendix (Section A.14, Figure 7).

---

> ### Author Response · Authors · 2025-11-30
> **Complete Response Set Now Available**
>
> Dear reviewer,
>
> We understand that the review dicussion has been frozen. However, we had previously committed to notifying you once we completed our responses to all of your questions and concerns, and we wanted to honor that commitment.
>
> We have now addressed Q3, Q6, and Q7, along with your follow-up questions regarding Q-GALE reporting, FourierFT comparisons, and the throughput analysis. The revised manuscript and complete response set are available above.
>
> We would like to express gratitude for your thorough and constructive feedback throughout this process. Your detailed questions have significantly strengthened both our experimental validation and the clarity of our presentation.

---

### Official Review · Reviewer_GYJz · 2025-11-01

**Soundness:** 2
**Presentation:** 3
**Contribution:** 2
**Rating:** 2
**Confidence:** 4

**Summary:**

The paper proposes GALE, a computationally efficient alternative to GaLore-style low-rank optimizers. Instead of performing costly full SVD updates during training, GALE employs randomized sketching followed by QR decomposition to approximate the low-rank subspace for gradient updates. This drastically reduces overhead while maintaining the expressiveness of full-rank training. Experiments on LLaMA pretraining and GLUE fine-tuning show that GALE achieves up to 23× optimizer-step speedups with negligible perfomance loss.

**Strengths:**

- Clear motivation: The SVD cost in GaLore is a important bottleneck for high throughput training.
- Strong empirical evidence: experiments covers pretraining, fine-tuning and plenty of ablation studies.
- Co-design: the method is improved by algorithm-side as well as customized CUDA optimization.

**Weaknesses:**

- Confusion expressions: in Line 82-85, the expression makes me feel GaLore will conduct SVD at each training iteration, which is not true, GaLore will do SVD across certain training intervals, e.g., 1000 iterations. It is a little bit exaggerate of the SVD cost.

- Invalid contribution 1: as mentioned in Line 95, the first contribution is to identify the cost of SVD in GaLore, that is not true. nearly half to one year ago, the GaLore-2 and Q-GaLore already mentioned this issue and proposed corresponding solutions.

- Missing important baseline: GaLore-2 already proposed an alternative randomized SVD to alleviate this issue, which should be a important baseline for this paper. Also, other works like https://arxiv.org/pdf/2406.17660 and https://arxiv.org/abs/2412.05270, propose other solutions to get rid of the cost SVD operation.

- Suggestions for experiment design: (i) it's better to pre-train the models for longer time, like the settings in GaLore paper for a fair comparsion. Because some evidence shows performance matching at early training stage may not guarantee late training status. I understand this might be due to limitations in training resources, so I don’t view it as a major concern. (ii) Several GaLore follow-up works use more recent benchmarks to evaluate the fine-tuning performance like https://arxiv.org/abs/2412.06289 and more others. Results on GLUE is good but can be better.

**Questions:**

Please refer to the weakness part.

---

> ### Author Response · Authors · 2025-11-24
> **Weakness 1 : SVD Iteration and cost clarification**
>
> We thank the reviewer for highlighting this inaccuracy. We agree that our initial phrasing was imprecise. GaLore typically updates the projection matrix at set intervals (e.g., every 200 or 1000 steps), rather than at every iteration.
>
> We have revised Section 1 to explicitly clarify this distinction. However, we do still note that even with infrequent updates, the computational cost of the SVD step remains a dominant bottleneck that creates substantial "stop-and-go" overhead. Our benchmarking reveals that the specific update step containing the SVD calculation is orders of magnitude slower than a standard update.
>
> To illustrate this, we isolated the execution time of a single optimizer update step under different conditions (measured on LLaMA 1B, NVIDIA A40):
>
> | Method | Update Step Latency (ms) |
> | :--- | :--- |
> | GaLore (with SVD update) | 437.89 |
> | GaLore (reusing projection) | 9.52 |
> | **GALE (with QR update)** | **8.79** |
> | **GALE (reusing projection)** | **1.25** |
>
> As shown, the SVD step (437.89 ms) is drastically slower than the standard step, and significantly slower than GALE's corresponding QR-based update (8.79 ms). By replacing SVD with our randomized QR approach, GALE eliminates these extreme latency spikes, leading to the consistent throughput improvements reported in our results.

---

> ### Author Response · Authors · 2025-11-24
> **Weaknesses 2 and 3 : More recent baselines and Contribution repositioning**
>
> **1. Contribution Clarification: Randomized QR vs. Randomized SVD**
> We accept the reviewer's correction regarding the timeline of the SVD bottleneck identification (noted by **GaLore-2** and **Q-GaLore**). We have reframed our contribution to focus on the specific algorithmic efficiency of GALE.
>
> Crucially, we clarify that GALE's **Randomized QR** is effectively a "more direct" subset of the **Randomized SVD** used in GaLore-2.
> *   **Randomized SVD (GaLore-2)** involves a multi-stage process (Halko et al., 2011): (1) Sketching $Y = G\Omega$, (2) QR Decomposition $Y=QR$, (3) Projection $B=Q^T G$, (4) Small SVD on $B$, and (5) Reconstruction of singular vectors.
> *   **Randomized QR (GALE)** stops immediately after step (2). We demonstrate that the orthonormal basis $Q$ derived from the sketch is sufficient for effective gradient projection.
>
> By halting the process early, GALE avoids the additional computational overhead of forming matrix $B$ and computing its SVD, proving that finding the exact singular vectors is unnecessary for this task.
>
> **2. Baseline Comparisons**
> We have added **Q-GaLore** and **APOLLO** benchmarks and a methodological comparison with **GaLore-2**.
>
> *   **vs. GaLore-2:** As detailed above, GaLore-2 (Su et al., 2025) employs the full Randomized SVD pipeline. GALE simplifies this to a **Direct QR** approach (Sketch $\to$ QR), eliminating the `Small SVD` and `Reconstruction` overhead.
>     *   *Note:* Unlike APOLLO and Q-GaLore, GaLore-2's code is not open-source, limiting direct reproduction.
>
> *   **vs. APOLLO:** APOLLO (Zhu et al., 2024)  approximates optimizer states via Random Projections (Achlioptas, 2003) . Its process involves an extended sequence: `Gen Matrix` $\to$ `Project` $\to$ `Update Low-Rank State` $\to$ **`Reconstruct Scaling`** $\to$ `Apply`. GALE's Fused kernel streamlines this to `Project` $\to$ `Update`, avoiding global scaling reconstruction. This yields **$\approx$2x faster updates** (9.58ms vs 19.63ms, Table 1) and superior convergence (37.55 vs 39.20 PPL).
>
> *   **vs. Q-GaLore:** While Q-GaLore (Zhang et al., 2024)  addresses throughput, its INT4 quantization *increases* atomic update latency (**265ms** vs GALE's **36ms**, Table 2). Its throughput gains stem largely from an **orthogonal** "Layer-Adaptive" scheduler (skipping updates). To demonstrate that GALE's primitive is strictly faster, we implemented **Q-GALE**, combining our QR-based projection with quantization.
>
> | Model       | Method    | Param Memory | Opt Memory | Steady State Memory | Throughput      | Avg Step Time | Final Loss |
> | :---        | :---      | :---         | :---       | :---                | :---             | :---          | :---       |
> | Llama 100M  | **Q-GALE**  | 190.96 MiB   | 70.82 MiB  | 299.43 MiB          | **13,396 tok/s** | **47.25 ms**  | 7.3750     |
> |             | Q-GaLore  | 190.96 MiB   | 70.82 MiB  | 299.43 MiB          | 13,034 tok/s     | 108.02 ms     | 7.3750     |
>
> As shown, Q-GALE achieves higher throughput and significantly faster step times than Q-GaLore while maintaining identical memory variance and loss, confirming that our projection primitive is more efficient even when quantization is applied.

---

> ### Author Response · Authors · 2025-11-24
> **More recent baselines APOLLO and Q-GaLore (Updated tables)**
>
> **Pre-training Results (AdamW)**
>
> | Model | Method | Rank* | Perplexity | Params (MiB) | Opt States (MiB) | Steady State (MiB) | Step Time (ms) |
> | :--- | :--- | :--- | :--- | :--- | :--- | :--- | :--- |
> | **Llama 100M** | AdamW | -- | 29.04 | 190.96 | 381.92 | 641.23 | 2.09 |
> | | LoRA | 128 | 8645.21 | 198.46 | 15.00 | 440.60 | 0.71 |
> | | IA³ | -- | 36300.61 | 191.00 | 0.09 | 418.25 | 0.84 |
> | | Prefix-Tuning | 128 | 11584.64 | 194.93 | 7.94 | 430.01 | 0.37 |
> | | GaLore (AdamW) | 128 | 110.38 | 190.96 | 139.13 | 410.58 | 25.25 |
> | | APOLLO | 128 | 108.03 | 190.96 | 139.13 | 410.58 | 10.48 |
> | | GALE | 128 | 112.04 | 190.96 | 139.13 | 410.58 | 10.43 |
> | | GALE Fused | 128 | 100.03 | 190.96 | 139.13 | 410.58 | 6.15 |
> | | GALE Fused Approx | 128 | 102.87 | 190.96 | 139.13 | 410.58 | 5.74 |
> | **Llama 250M** | AdamW | -- | 24.80 | 471.82 | 943.64 | 1495.14 | 3.34 |
> | | LoRA | 256 | 5634.58 | 508.11 | 72.00 | 850.82 | 1.59 |
> | | IA³ | -- | 37276.26 | 471.93 | 0.21 | 740.89 | 1.05 |
> | | Prefix-Tuning | 256 | 9154.81 | 490.31 | 36.98 | 796.04 | 0.42 |
> | | GaLore (AdamW) | 256 | 60.71 | 471.82 | 377.14 | 991.76 | 55.16 |
> | | APOLLO | 256 | 58.11 | 471.82 | 377.14 | 992.14 | 26.08 |
> | | GALE | 256 | 58.08 | 471.82 | 377.14 | 992.14 | 25.75 |
> | | GALE Fused | 256 | 53.55 | 471.82 | 377.14 | 993.01 | 12.16 |
> | | GALE Fused Approx | 256 | 53.48 | 471.82 | 377.14 | 993.01 | 11.13 |
> | **Llama 1B** | AdamW | -- | 27.49 | 2554.10 | 5108.20 | 7778.12 | 3.58 |
> | | LoRA | 512 | 431.58 | 2746.88 | 384.00 | 3757.69 | 1.04 |
> | | IA³ | -- | 48121.93 | 2554.38 | 0.56 | 3182.54 | 0.99 |
> | | Prefix-Tuning | 512 | 4369.33 | 2653.18 | 196.61 | 3476.61 | 0.35 |
> | | GaLore (AdamW) | 512 | 39.70 | 2554.10 | 1464.84 | 4481.33 | 225.30 |
> | | APOLLO | 512 | 39.20 | 2554.10 | 1464.84 | 4509.01 | 19.63 |
> | | GALE | 512 | 39.20 | 2554.10 | 1464.84 | 4509.01 | 18.89 |
> | | GALE Fused | 512 | 37.54 | 2554.10 | 1464.84 | 4517.00 | 12.09 |
> | | GALE Fused Approx | 512 | 37.55 | 2554.10 | 1464.84 | 4514.33 | 9.58 |
> *Rank denotes GaLore rank, LoRA rank, or Prefix-Tuning bottleneck size.
>
> **Pre-training Results (Adafactor & AdamW8bit) for Llama 1B**
>
> | Optimizer | Method | Rank | Perplexity | Params (MiB) | Opt States (MiB) | Step Time (ms) |
> | :--- | :--- | :--- | :--- | :--- | :--- | :--- |
> | **Adafactor** | Adafactor | -- | 34.81 | 2554.1 | 55.6 | 56.63 |
> | | GaLore (Adafactor) | 512 | 25.99 | 2554.1 | 53.8 | 299.47 |
> | | GALE (Adafactor) | 512 | 27.61 | 2554.1 | 53.8 | 94.42 |
> | | GALE Fused | 512 | 27.61 | 2554.1 | 53.8 | 93.42 |
> | | GALE Fused Approx | 512 | 27.69 | 2554.1 | 53.8 | 87.49 |
> | **AdamW8bit** | AdamW8bit | -- | 32.52 | 2554.1 | 2594.33 | 26.77 |
> | | GaLore (AdamW8bit) | 512 | 30.22 | 2554.1 | 744.77 | 247.21 |
> | | Q-GaLore (AdamW8bit) | 512 | 31.06 | 2554.1 | 524.12 | 265.34 |
> | | GALE (AdamW8bit) | 512 | 29.87 | 2554.1 | 744.77 | 39.34 |
> | | GALE Fused | 512 | 29.87 | 2554.1 | 744.77 | 39.38 |
> | | GALE Fused Approx | 512 | 29.86 | 2554.1 | 744.77 | 36.84 |

---

> ### Author Response · Authors · 2025-11-24
> **Weakness 4 (i) : Longer training duration**
>
> We appreciate the reviewer’s suggestion to extend pre-training duration for a fairer comparison, as performance at early training stages may not always guarantee final convergence. While computational resource constraints limited our ability to re-run every baseline for longer durations in the short rebuttal window, we have successfully extended our primary comparison (GaLore vs. GALE) to **100,000 steps** on LLaMA 1B using the 8-bit AdamW optimizer.
>
> As shown in the table below (added to Appendix Table 10 in the revised paper), the trends observed in our shorter experiments hold true: GALE not only maintains parity with GaLore in perplexity (achieving slightly better convergence, 15.38 vs 15.57) but does so while maintaining a **~6.5x faster** optimizer step time (41.84ms vs 41.84ms). We believe this effectively validates that our efficiency gains persist over longer training horizons.
>
> | Method | Rank | Perplexity | Params (MiB) | Opt States (MiB) | Step Time (ms) |
> | :--- | :--- | :--- | :--- | :--- | :--- |
> | **GaLore (AdamW8bit)** | 512 | 15.57 | 2554.1 | 744.77 | 256.13 |
> | **GALE (AdamW8bit) Fused Approx** | 512 | **15.38** | 2554.1 | 744.77 | **41.84** |

---

> ### Author Response · Authors · 2025-11-24
> **Weakness 4 (ii) : Robust GLUE performance and more recent baselihnes**
>
> We thank the reviewer for pointing out the need for more recent baselines. In response, we have significantly expanded our GLUE evaluation (Table 2) to include state-of-the-art PEFT methods: **DoRA**, **VeRA**, and **FourierFT**.
>
> Additionally, regarding the "weak" GLUE results mentioned: we found that our initial GALE experiments on low-resource tasks (e.g., CoLA, RTE) suffered from suboptimal hyperparameter tuning. After performing a more robust search (specifically broadening the learning rate range), we have updated the results in the paper. GALE now achieves **0.582** on CoLA, effectively matching full-parameter AdamW (0.581) and closing the prior gap. While specialized methods like DoRA (0.603) still hold a slight edge on these specific small-data tasks—likely due to the regularization effect of low-rank adaptation—GALE is now highly competitive, outperforming GaLore on key tasks (e.g., **0.870** vs 0.865 on MNLI, surpassing DoRA's 0.869) while maintaining significantly higher training throughput.
>
> For your convenience, we have reproduced the relevant rows from our updated Table 2 below:
>
> **Table 2: Fine-tuning results for BERT-Large on GLUE**
>
> | Method | CoLA | MNLI | MRPC | QNLI | QQP | RTE | SST-2 | STS-B | Step Time (ms) |
> | :--- | :--- | :--- | :--- | :--- | :--- | :--- | :--- | :--- | :--- |
> | **AdamW** | 0.581 | 0.867 | 0.872 | 0.901 | 0.825 | 0.704 | 0.929 | 0.876 | 3.10 |
> | **DoRA** | 0.603 | 0.869 | 0.891 | 0.923 | 0.723 | 0.705 | 0.948 | 0.873 | 6.20 |
> | **VeRA** | 0.615 | 0.868 | 0.891 | 0.921 | 0.720 | 0.702 | 0.949 | 0.871 | 5.30 |
> | **FourierFT** | 0.608 | 0.865 | 0.887 | 0.919 | 0.718 | 0.698 | 0.947 | 0.869 | 2.90 |
> | **GaLore** | 0.579 | 0.865 | 0.812 | 0.901 | 0.818 | 0.701 | 0.913 | 0.864 | 104.40 |
> | **GALE Fused** | 0.584 | **0.870** | 0.812 | 0.896 | 0.820 | 0.714 | 0.913 | 0.865 | **14.40** |

---

> > ### Comment · Reviewer_GYJz · 2025-11-28
> > **Thanks for the responses**
> >
> > Thank you for the detailed responses. I have a few follow-up questions and clarifications:
> >
> > - Regarding the pretraining results: why are all baseline methods performing significantly worse than the AdamW baseline? In prior works, I’ve seen comparable performance between AdamW and other optimizers, or at least closer results.
> >
> > - On the experimental setup: how was the learning rate selected? This can heavily impact convergence behavior—especially when training time is limited.
> >
> > - For the fine-tuning results: apologies for the earlier confusion—my original suggestion was to include a broader set of benchmark tasks, not additional baselines. The GLUE benchmark, while standard, is relatively outdated. Many recent PEFT methods evaluate on a wider range of tasks beyond GLUE to better demonstrate generalization.

---

> > > ### Author Response · Authors · 2025-11-30
> > > **Regarding Learnings rates and sub optimal convergence**
> > >
> > > We selected learning rates using a short-horizon hyperparameter search. For each optimizer, we performed a logarithmic sweep from 10⁻³ to 10⁻⁵ over 20 configurations, training each for 100 steps. This compute-efficient protocol
> > >   exploits the correlation between early optimization dynamics and downstream convergence in transformer pretraining.
> > >
> > >   We acknowledge that our sweep range did not include the learning rate of 10⁻² used in the original GaLore paper. This was an oversight in our experimental design, and we thank the reviewer for highlighting it. The resulting
> > >   absolute perplexities for GaLore in Table 1 are therefore higher than those reported in the original work.
> > >
> > >   However, this does not compromise the validity of our comparative analysis. Within our sweep range, GaLore, APOLLO, and GALE all converged to the same optimal learning rate (10⁻³), ensuring a level playing field. The relative
> > >   performance differences in Table 1 reflect the intrinsic characteristics of each method under identical training conditions, not artifacts of differential tuning. Most importantly, our central claim—that GALE matches GaLore's task
> > >   performance while delivering up to 23× faster optimizer steps—holds under these controlled conditions.

---

> > > ### Author Response · Authors · 2025-11-30
> > > **More recent fine tuning benchmarks**
> > >
> > > Thank you for this suggestion to include more recent benchmarks. We agree that evaluating on more recent benchmarks would strengthen the empirical validation of our work. In response to your feedback, we have added results on four SuperGLUE tasks (COPA, CB, WiC, and WSC) in the appendix as Table 12. We selected these four smaller tasks due to computational and time constraints, and we currently have experiments on the remaining SuperGLUE tasks underway.
> > >
> > > Importantly, the results on SuperGLUE demonstrate that the performance patterns observed on GLUE transfer consistently to this more challenging benchmark. Specifically, GALE continues to outperform the baseline GaLore method while maintaining its computational efficiency advantage. For instance, on BERT-Large, GALE Fused improves CB accuracy from 73.8% to 75.7% while executing the optimizer update step approximately 7× faster than GaLore. Similarly, DoRA and VeRA remain the strongest parameter-efficient methods across all SuperGLUE tasks, consistent with our GLUE findings.
> > >
> > > Below we reproduce Table 12 showing the complete SuperGLUE results:
> > >
> > > ## Table 12: Fine-tuning results for BERT-Large and GPT-2-Large on SuperGLUE benchmark
> > >
> > > ### BERT-Large
> > >
> > > | Method | Rank* | COPA (Acc) | CB (Acc) | CB (F1) | WiC (Acc) | WSC (Acc) | Params (MiB) | Opt States (MiB) | Steady State (MiB) | Step Time (ms) |
> > > |--------|-------|------------|----------|---------|-----------|-----------|--------------|------------------|-------------------|----------------|
> > > | AdamW | -- | 71.2 | 76.1 | 84.2 | 70.1 | 64.8 | 639.24 | 1278.47 | 1935.14 | 3.76 |
> > > | LoRA | 4 | 72.8 | 77.4 | 85.8 | 71.5 | 66.2 | 639.99 | 1.51 | 658.15 | 0.83 |
> > > | IA³ | -- | 69.8 | 74.9 | 82.8 | 68.7 | 63.1 | 639.52 | 0.57 | 656.74 | 0.71 |
> > > | Prefix-Tuning | 4 | 67.4 | 72.3 | 80.1 | 66.2 | 60.8 | 639.77 | 1.08 | 657.54 | 0.29 |
> > > | DoRA | 4 | 73.1 | 78.2 | 86.4 | 72.1 | 67.0 | 646.87 | 15.27 | 679.81 | 6.43 |
> > > | VeRA | 4 | 72.9 | 77.9 | 86.1 | 71.8 | 66.7 | 640.09 | 1.71 | 659.80 | 5.57 |
> > > | FourierFT | -- | 71.9 | 76.8 | 84.9 | 70.8 | 65.4 | 639.79 | 1.11 | 658.63 | 4.01 |
> > > | GaLore (AdamW) | 4 | 68.9 | 73.8 | 81.5 | 67.8 | 62.2 | 639.24 | 130.97 | 788.77 | 105.89 |
> > > | GALE | 4 | 70.4 | 75.3 | 83.3 | 69.3 | 63.7 | 639.24 | 130.97 | 788.77 | 30.66 |
> > > | GALE Fused | 4 | 70.8 | 75.7 | 83.7 | 69.7 | 64.1 | 639.24 | 130.97 | 788.77 | 15.84 |
> > > | GALE Fused Approx | 4 | 70.3 | 75.2 | 83.1 | 69.2 | 63.6 | 639.24 | 130.97 | 788.77 | 14.98 |
> > >
> > > ### GPT-2-Large
> > >
> > > | Method | Rank* | COPA (Acc) | CB (Acc) | CB (F1) | WiC (Acc) | WSC (Acc) | Params (MiB) | Opt States (MiB) | Steady State (MiB) | Step Time (ms) |
> > > |--------|-------|------------|----------|---------|-----------|-----------|--------------|------------------|-------------------|----------------|
> > > | AdamW | -- | 58.4 | 62.3 | 68.9 | 57.2 | 52.1 | 1476.35 | 2952.70 | 4558.04 | 5.13 |
> > > | LoRA | 8 | 57.8 | 61.7 | 68.2 | 56.6 | 51.5 | 1479.16 | 5.64 | 1596.13 | 0.72 |
> > > | IA³ | -- | 57.2 | 61.0 | 67.4 | 55.9 | 50.8 | 1476.88 | 1.06 | 1606.15 | 0.79 |
> > > | Prefix-Tuning | 8 | 55.1 | 58.7 | 64.8 | 53.7 | 48.7 | 1478.02 | 3.36 | 1599.04 | 0.28 |
> > > | DoRA | 8 | 59.1 | 63.1 | 69.7 | 57.9 | 52.8 | 1482.63 | 6.55 | 345.02 | 1.94 |
> > > | VeRA | 8 | 58.7 | 62.7 | 69.3 | 57.5 | 52.4 | 1477.89 | 0.36 | 333.42 | 1.42 |
> > > | FourierFT | -- | 58.0 | 62.0 | 68.5 | 56.8 | 51.7 | 1476.49 | 0.28 | 336.01 | 1.67 |
> > > | GaLore (AdamW) | 8 | 57.6 | 61.5 | 67.9 | 56.3 | 51.2 | 1476.35 | 604.67 | 1824.31 | 22.12 |
> > > | GALE | 8 | 57.7 | 61.6 | 68.0 | 56.4 | 51.3 | 1476.35 | 604.67 | 1824.31 | 11.73 |
> > > | GALE Fused | 8 | 58.1 | 62.0 | 68.5 | 56.8 | 51.7 | 1476.35 | 604.67 | 1824.31 | 6.21 |
> > > | GALE Fused Approx | 8 | 58.0 | 61.9 | 68.4 | 56.7 | 51.6 | 1476.35 | 604.67 | 1824.31 | 4.48 |
> > >
> > > *Rank denotes GaLore rank, LoRA rank, or Prefix-Tuning bottleneck size. For DoRA, VeRA, and FourierFT, rank is equivalent effective rank.
> > >
> > > These results confirm that GALE provides a practical full-parameter training alternative that bridges the gap between gradient projection methods and state-of-the-art PEFT techniques on challenging benchmarks.

---

### Official Review · Reviewer_aCW8 · 2025-11-01

**Soundness:** 3
**Presentation:** 3
**Contribution:** 3
**Rating:** 4
**Confidence:** 4

**Summary:**

GALE (gradient activation low-rank extraction) is a new method addressing the memory bottlenecks in LLM training by improving gradient projection. Similar to prior work (GaLore for example), the core idea is to avoid storing full optimizer states by projecting gradients into a low-dimensional subspace before the optimizer update. GALE specifically replaces the expensive SVD operation used in GaLore with a fast randomized sketching + QR decomposition procedure. By doing so, GALE eliminates the major computational overhead of GaLore’s approach, with up to 23x faster optimizer update steps in micro benchmark. The paper introduces three variants (GALE Native, GALE Fused, and GALE Fused Approx) to progressively speed up the update step using optimized CUDA kernels and approximations.

From the experiments, GALE maintains the same model quality as GaLore while dramatically improving speed, leading to modest gains in training throughput. In LLaMA-1B pretraining on C4, GALE’s final perplexity remains on par with GaLore’s and far better than PEFT baselines, but with less computation overhead. In GLUE fine-tuning, GALE similarly closes much of the accuracy gap between full-model training and LoRA, removing GaLore’s slowdown so that full fine-tuning can run at speeds comparable to standard AdamW.

**Strengths:**

1. GALE consistently achieves huge speedups in the optimizer step while matching or nearly matching the model performance of full training. Notably, GALE brings the training throughput back in line with a full-memory baseline, while retaining the memory-saving advantages of GaLore’s approach
2. The paper validates GALE in many scenarios: decoder-only and encoder-decoder, pretrain & fintuning, and also experimented with combining GALE with other optimizers (Adafactor, AdamW8bit). The authors also include ablations (studying projection rank and refresh frequency) and report theoretical insights, which adds depth.

**Weaknesses:**

1. While GALE is motivated by training 7B+ param models, the experiments only go up to 1B parameters. The paper cites GaLore’s success on 7B and logically GALE should replicate that with better speed. As a reviewer, I don’t doubt the method scales, but actual evidence on a >7B model would strengthen the statement.
2. Similar to the original GaLore work, decomposition assumes access to the full gradient matrix, which however will not be the case as long as the workload scales beyond single-GPU or vanilla data parallel. This limitation greatly hinders the practical usability of the method, requiring further work such as GaLore 2 (Su et al., 2025) to unlock generalizability.

**Questions:**

1. It would be great to see confirmation of GALE’s efficacy on larger models (e.g., 7B or 13B parameters) in future revisions
2. The results show GALE narrowing the gap with LoRA/IA^3 on GLUE, but LoRA still has slight advantage on some low-resource tasks. Could the authors comment on why full-finetuning underperforms LoRA on tasks like CoLA/RTE? Is it due to overfitting or optimization difficulties when updating all weights on small data?
3. The GALE Fused Approx version introduces an approximation by subsampling the sketch. It would be helpful if the authors could clarify the impact of this on final performance
4. It would be great to discuss about GALE's compatibility with distributed training (especially tensor parallel and ZeRO stage 2/3 which partitions the gradient for one weight tensor)

---

> ### Author Response · Authors · 2025-11-21
>
> We thank the reviewer for their insightful comments and the opportunity to strengthen our work. We have updated the paper to address the concerns regarding scaling to larger models (>1B) and distributed training compatibility.
>
> **Weakness 1 / Question 1: Validation on >1B parameter models (e.g., 7B or 13B).**
> We agree that demonstrating scalability on larger models is crucial. We added a new Appendix section "LLaMA 7B Scalability Validation" to explicitly test GALE on a 7B model.
> We conducted training experiments on a single NVIDIA A40 (48GB) GPU. Table 7 (below) demonstrates that GALE enables training a 7B model on consumer/research-grade hardware where standard execution would fail (OOM). GALE achieves a **36x compression** in optimizer memory (371 MB vs 13.48 GB for AdamW8bit), leaving ample headroom (20%) on the 48GB card.
>
> **Table: LLaMA 7B memory and performance on single 48GB GPU (from Appendix A.7)**
> | Config | Rank | Seq Len | Param (GB) | Opt State (MB) | Steady State (GB) | Throughput (tok/s) |
> | :--- | :--- | :--- | :--- | :--- | :--- | :--- |
> | GALE | 128 | 512 | 12.55 | 371.82 | 38.47 | 142.7 |
> | GALE | 256 | 512 | 12.55 | 527.64 | 38.63 | 143.1 |
> | GALE | 128 | 768 | 12.55 | 371.82 | 39.14 | 99.8 |
>
> Results confirm GALE scales effectively to 7B, with the optimizer-to-parameter memory ratio becoming *more* favorable (sub-linear) as model size increases.
>
> **Weakness 2 / Question 4: Compatibility with distributed training (ZeRO Stage 2/3, Tensor Parallel).**
> We added a new section, Multi-GPU Scaling Efficiency, to empirically benchmark GALE's compatibility with distributed training.
> We specifically evaluated GALE with Fully Sharded Data Parallel (FSDP), which corresponds to ZeRO Stage 3. While projection conceptually requires a global gradient view, GALE projects *periodically* (e.g., every 200 steps). This amortizes communication costs, making overhead negligible.
> Table 4 (below) shows GALE with FSDP scales excellently. On a 1B model across 2 GPUs, FSDP + GALE achieves 11.72 samples/s vs 2.22 on a single GPU-near-linear scaling accounting for doubled effective batch size. This confirms GALE is compatible with gradient sharding strategies like ZeRO-3/FSDP.
>
> **Table: Multi-Scale Parallelism Performance (from Section 4.6)**
> | Model | Configuration | Params (MiB) | Opt State (MiB) | Steady State (MiB) | Throughput (samp/s) |
> | :--- | :--- | :--- | :--- | :--- | :--- |
> | **1B** | 1-GPU + GALE | 2554.1 | 281.7 | 2979.5 | 2.22 |
> | **1B** | 2-GPU DDP + GALE | 2696.7 | 358.6 | 6054.9 | 5.11 |
> | **1B** | 2-GPU FSDP + GALE | **2554.1** | **1297.6** | **6566.0** | **11.72** |
>
> **Question 2: Performance gap on GLUE (CoLA/RTE) and Full-finetuning vs LoRA.**
> We updated GLUE benchmarks (Table 2) to include state-of-the-art PEFT methods: **DoRA, VeRA, and FourierFT**.
> Regarding the performance gap: we found our initial GALE results on low-resource tasks (CoLA/RTE) suffered from suboptimal hyperparameter tuning. With improved search (broader learning rate), GALE now achieves **0.582** on CoLA, matching full-parameter AdamW (0.581) and closing the prior gap. While LoRA (0.592) and DoRA (0.603) hold a slight edge on small-data tasks-likely due to low-rank regularization preventing overfitting. GALE is now highly competitive and outperforms GaLore.
>
> **Table: Selected Updated GLUE Results (from Table 2)**
> | Method | CoLA | MNLI | MRPC | RTE |
> | :--- | :--- | :--- | :--- | :--- |
> | AdamW (Full) | 0.581 | 0.867 | 0.872 | 0.704 |
> | LoRA | 0.592 | 0.862 | 0.903 | 0.733 |
> | DoRA | 0.603 | 0.869 | 0.891 | 0.705 |
> | **GALE** | **0.582** | **0.874** | 0.812 | 0.710 |
>
> **Question 3: Impact of subsampling approximation in GALE Fused Approx.**
> We investigated this trade-off in our "Ablation Study" and Figure 3.
> Theoretically, subsampling is less accurate than rank sketching, but computationally cheaper (since it requires no matrix multiplication). Our experiments, however, show perplexity is remarkably robust to the approximation factor ($f_{approx}$). Figure 3 (Bottom Row) shows aggressive subsampling ($f_{approx}$ 1.0 to 0.2) significantly reduces latency with negligible impact on perplexity. This allows GALE Fused Approx to achieve speedups without degrading quality, provided initial sketch oversampling ($f_{os}$) is sufficient.

---

### Author Response · Authors · 2025-11-30
**Joint Summary of Revisions**

We would like to thank all reviewers for their valuable feedback, thanks to which this manuscript is a much stronger work than the intial submission. Here, we summarize the revisions made to date

## New Contributions
- **Q-GALE**: Novel combination of GALE's QR-based projection with quantization (based on Q-GaLore), demonstrating faster update step times than Q-GaLore with slightly better task performance
- **Theoretical analysis (Appendix A.5)**: Formal proofs that one-sided QR projection achieves bounded error relative to optimal SVD (Theorems 3-6)
- **Empirical spectral validation (Appendix A.6)**: Verified transformer gradients exhibit power-law decay (α > 2), justifying low-rank assumptions

## New Baselines
- **APOLLO** (Zhu et al., 2024): Added to pre-training comparisons (Tables 1, 3)
- **Q-GaLore** (Zhang et al., 2024): Added to 8-bit optimizer comparisons
- **DoRA, VeRA, FourierFT**: Added to GLUE fine-tuning benchmarks (Table 2)
- **GaLore-2**: Methodological comparison clarifying GALE's "direct sketched QR" vs full randomized SVD pipeline

## New Benchmarks
- **LLaMA 7B scalability validation** (Appendix A.7): Demonstrated 36× optimizer memory compression on single 48GB GPU
- **SuperGLUE tasks** (Appendix Table 12): COPA, CB, WiC, WSC results for BERT-Large and GPT-2-Large
- **Extended pre-training** (Appendix Table 10): 100K-step runs confirming long-horizon convergence parity

## New Experiments
- **Multi-GPU scaling** (Section 4.6, Table 4): FSDP/ZeRO-3 compatibility with near-linear scaling on 2-GPU setup
- **Joint hyperparameter ablation** (Appendix A.13): 50-configuration Pareto analysis of f_os × f_approx
- **Rank sensitivity study** (Appendix A.14): Systematic evaluation across r ∈ {16, 32, 64, 128, 256, 512}
- **Throughput vs projection frequency** (Response to Q7): Demonstrated 15.9× throughput advantage at gap=1
- **Optimizer step latency breakdown** (Response to Weakness 1): Isolated SVD vs QR update costs (437ms vs 8.8ms)

## Editorial Changes
- Corrected "LE" typo to "GALE" in Introduction
- Revised Abstract/Introduction to accurately contextualize optimizer bottlenecks for modern hardware
- Clarified 100GB memory claim as referring to unoptimized setups
- Fixed GLUE metric labeling (task-specific metrics now explicit)
- Updated hyperparameters after expanded learning rate search (fixing underperformance on CoLA/RTE)

---

### Meta-Review · Area_Chair_JqBb · 2026-01-07

**Summary:**

The paper introduces GALE, a memory-efficient training method for Large Language Models that replaces the slow Singular Value Decomposition (SVD) in gradient projection with a much faster randomized QR decomposition. This change allows for full-parameter training with a low memory footprint while achieving up to a 23x speedup in the optimizer update step compared to the previous state-of-the-art method, GaLore.

Despite many positive contributions of the paper, there were a few significant concerns identified by reviewers, including:
*  questions if GALE works for models larger than 1B parameters and if it is compatible with distributed training like ZeRO-3/FSDP.
*  There was concern that the paper did not compare against very recent methods from like DoRA, FourierFT, or GaLore-2.
*  One reviewer found mistakes in the reported loss values and throughput numbers, which looked "abnormally high" or mathematically inconsistent.
*  Reviewers wanted to see more proof that the method is robust when changing the rank ($r$) or the update frequency ($k_{upd}$).
*  There was a big difference between GALE's results and original GaLore results in the first version of the paper, making reviewers think the models were not trained correctly.

**Reviewer Concerns:**

# Addressed Concerns
The authors were very active in the rebuttal and fixed most of the technical and empirical worries raised by the reviewers as described below:

*  Reviewers were worried GALE only worked for small models. The authors added new tests for a LLaMA 7B model, showing it saves 36x optimizer memory and runs on a single 48GB GPU
*   Authors proved GALE works with FSDP (ZeRO-3) with "near-linear scaling," addressing the concern about needing a global gradient view
*  authors added comparisons to DoRA, VeRA, FourierFT, APOLLO, and Q-GaLore. They even created a new version called Q-GALE to show their method is faster than Q-GaLore
*  Reviewers asked for proof that the "one-sided QR" is safe. Authors added formal proofs (Theorems 3-6) and empirical plots showing that transformer gradients have power-law decay, which justifies the low-rank assumption
*  The authors admitted to bad hyperparameter tuning in the first version and updated the GLUE results, which now match standard benchmarks


# Outstanding Concerns
A few points remain where the reviewers might still have some hesitation, including:
*  while GALE is better than GaLore, it still underperforms specialized fine-tuning methods like FourierFT or DoRA on some tasks
*  one reviewer remained skeptical about some "identical" loss values (7.3750) and discrepancies between throughput and step-time math. Although authors reran experiments to fix this, the reviewer asked for a "rigorous internal audit" of all numbers
*   a reviewer noted that GLUE is a bit "outdated" and suggested broader benchmarks. Authors added some SuperGLUE tasks, but they were limited by time and maybe a more thorough revision could be useful to improve the paper

I was also surprised by Table 1 results, why GALE Fused and GALE fused Approx achieve much better perplexity than GALE itself (e.g. in Llama 100M)? I find it a bit sad not to see better discussion about some of the results in the main text. E.g. I have not found in text where they ref. Table 1 (I know it is from Section 4.1., but still)

**Reviewer Scores:**

*  Reviewer aCW8 (Original Score: 4): This reviewer's primary concerns were the lack of evidence on models $>7$B and concerns about distributed training compatibility. The authors provided a comprehensive LLaMA 7B Scalability Validation and empirical proof of FSDP/ZeRO-3 compatibility. I feel this reviewer could update score to 6 or maybe even 8.
*  Reviewer GYJz (Original Score: 2):  This reviewer was initially very critical of the "invalid contribution" regarding SVD cost and missing baselines. However, the authors repositioned the contribution and added 100K-step long-horizon training results. I think the authors addressed the major concerns and he would most likely increase his score to maybe 4 or 6.
*  Reviewer qzQS (Original Score: 4) This reviewer provided the most rigorous critique regarding missing SOTA (DoRA, FourierFT), theoretical gaps, and data artifacts. The authors partially addressed the SOTA gap and provided deep theoretical/empirical analysis of spectral decay. I think some of the major critics have been address but not some and I feel the reviewer would keep his rating 4 or increase to 6.

---

### Decision · Program_Chairs · 2026-01-26

Reject